# An automated fracture trace detection technique using the complex shearlet transform

Rahul Prabhakaran[1,2], Pierre-Olivier Bruna[1], Giovanni Bertotti[1], and David Smeulders[2]

[1]Department of Geoscience and Engineering, Delft University of Technology, Delft, the Netherlands
[2]Department of Mechanical Engineering, Eindhoven University of Technology, the Netherlands

**Correspondence:** Rahul Prabhakaran (r.prabhakaran@tudelft.nl / r.prabhakaran@tue.nl )

**Abstract.** Representing fractures explicitly using a discrete fracture network (DFN) approach is often necessary to model the complex physics that govern thermo-hydro-mechanical-chemical processes (THMC) in porous media. DFNs find applications in modelling geothermal heat recovery, hydrocarbon exploitation, and groundwater flow. It is advantageous to construct DFNs from photogrammetry of fractured outcrop analogues as the DFNs would capture realistic, fracture network properties. Recent advances in drone photogrammetry have greatly simplified the process of acquiring outcrop images, and there is a remarkable increase in the volume of image data that can be routinely generated. However, manually digitizing fracture traces is time-consuming and inevitably subject to interpreter bias. Additionally, variations in interpretation style can result in different fracture network geometries, which, may then influence modelling results depending on the use-case of the fracture study. In this paper, an automated fracture trace detection technique is introduced. The method consists of ridge detection using the complex shearlet transform coupled with post-processing algorithms that threshold, skeletonize, and vectorize fracture traces. The technique is applied to the task of automatic trace extraction at varying scales of rock discontinuities, ranging from $10^0$-$10^2$ m. We present automatic trace extraction results from three different fractured outcrop settings. The results indicate that the automated approach enables extraction of fracture patterns at a volume beyond what is manually feasible. Comparative analysis of automatically extracted results with manual interpretations demonstrates that the method can eliminate the subjectivity that is typically associated with manual interpretation. The proposed method augments the process of characterizing rock fractures from outcrops.

## 1 Introduction

Naturally fractured reservoir (NFR) modelling requires an explicit definition of fracture network geometry to accurately capture the effects of fractures on the overall reservoir behaviour. The National Research Council (1996) suggested the idea of using geologically realistic outcrop fracture patterns to guide subsurface fracture modelling. In recent work, the use of deterministic discrete fracture networks (DFNs) based on trace digitization from photogrammetry of outcrop analogues was investigated by Bisdom et al. (2017) and Aljuboori et al. (2015) for reservoir fluid flow simulation and well testing. Outcrop derived DFNs encapsulate 2D fracture network properties at a scale that cannot be characterized using either standard surface approaches (scanlines and satellite imagery) or subsurface techniques (seismic imaging/borehole imagery/core sampling). Ukar

et al. (2019) suggested a comprehensive set of protocols to select fractured outcrops that are representative of the subsurface. Stochastic and geomechanical DFNs are alternatives to outcrop derived DFNs for fractured reservoir modeling. Stochastically generated DFNs have the disadvantage that they cannot replicate the spatial organization of fracture network patterns observed in nature (Thovert et al., 2017). Geomechanically derived DFNs are based on the physics of fracture propagation (for e.g.

5   Olson et al., 2009; Thomas et al., 2018) and can reproduce realistic fracture patterns provided the complex paleostress field and paleo rock properties are known; however, they are also computationally intensive and hence have limited applicability. A carefully chosen fractured outcrop that is relatively free of noise (fractures resulting from exhumation and weathering and not too much hidden by vegetation) may be used to interpret realistic fracture networks which are geometrical inputs used in simulating various subsurface thermo-hydro-mechanical-chemical (THMC) processes.

Recent advances in Unmanned Aerial Vehicles (UAVs) and stereo-photogrammetry has dramatically simplified the acquisition of georeferenced datasets of fractured outcrop images (for e.g. Bemis et al., 2014; Harwin and Lucieer, 2012; Turner et al., 2012). Photogrammetry using the Structure from Motion (SfM) principle is a relatively inexpensive and rapid technique by which 3D outcrop models are built by identifying, extracting, and positioning common points in georeferenced

outcrop images (Donovan and Lebaron, 2009). Images are captured using a camera-equipped UAV that is capable of following pre-programmed flight missions where flight path, altitude, velocity, and overlap are specified. The images undergo further processing steps that include generating sparse point clouds of common points, aligning the images, generating dense point clouds (3D representation of outcrop geometry), and generating meshed surfaces (Bisdom et al., 2017). Interpreting fractures on the image orthomosaics with conventional Geographic Information System (GIS) software completes the outcrop-based

DFN workflow.

Manually interpreting fractures is time-consuming and forms a bottleneck in an outcrop-based DFN workflow. A manual interpretation has a fair degree of associated subjectivity, and interpreter bias may take the form of specific scales of features being inadvertently omitted or deliberately ignored (Bond et al., 2007; Scheiber et al., 2015). Manual interpretation also suffers

from a lack of repeatability owing to the level of expertise of the interpreter, and the interpretation criteria followed (Hillier et al., 2015; Sander et al., 1997). Reproducibility may not be guaranteed even with the same interpreter in multiple trials (Mabee et al., 1994). According to Bond et al. (2015), quantifying the magnitude and impact of subjective uncertainty is difficult. Long et al. (2018) conducted a study on variability of fracture interpretation in which geologists with varying levels of expertise interpreted a single image. They found considerable variation in fracture topology, orientation, intensity, and length distributions

in the interpretations. Andrews et al. (2019) made a detailed quantification of subjective bias in scanlined-based fracture data collection, the associated effects on derived fracture statistics and suggested protocols for managing the variations. Peacock et al. (2019) delved into the multiple reasons for bias and the resulting implications for modelling. Given the amount of data generated in short UAV flight missions, man-hours spent in interpretation, and the need to de-bias interpretation as much as possible, automatic feature detection techniques may be considered. Automated approaches can speed up the process, improve

accuracy, and exploit the acquired data to the fullest possible extent.

In this paper, we apply an automated method to extract digitized fracture traces from images of fractured rocks. The method utilizes the complex shearlet transform measure to extract fracture ridge realizations from images. Post-processing image analysis algorithms are coupled with the ridge extraction process to vectorize fracture traces in an automated manner. The complex shearlet transform was introduced by Reisenhofer (2014) and King et al. (2015) and previously applied to problems such as detecting coastlines from Synthetic Aperture Radar (SAR) images (King et al., 2015) and propagating flame fronts from planar laser-induced fluorescence (PLIF) images (Reisenhofer et al., 2016). We present automatic fracture extraction results from drone images of two carbonate outcrops (Parmelan, France and Brejões, Brazil) and station scale images of igneous dyke swarms.

## 2  Background

### 2.1  Review of automated and semi-automated fracture detection approaches

Rapid digitization of geological features from photogrammetry is challenging owing to issues like spatially varying image resolution, inadequate exposure, the presence of shadows due to effects of topography on illumination conditions, and chromatic variations of essential features. False positives are non-geological features (such as trees, shrubbery, and human-made structures) that are detected using semi-automated / automated approaches (Vasuki et al., 2014). Removal of false positives is time-consuming. On the other hand, essential features that are not detected at all (referred to as false negatives) by an algorithm, further complicates the task of automated feature extraction. Automated methods, in general, detect more features than what is present in the image (Abdullah et al., 2013). In this section, we review some approaches for automatic fracture detection based on the class of algorithm used.

Automated fracture detection utilizing higher dimensional data such as point clouds, digital elevation models (DEMs) and digital terrain models (DTMs) have an advantage in that depth variations are captured and can be used to extract features. Thiele et al. (2017) presented an approach based on a least cost function algorithm applicable to ortho-photographs of jointed fracture sets and 3D point cloud data. Masoud and Koike (2017) introduced a software package to detect lineaments from composite grids derived from gravity, magnetic, DEMs, and satellite imagery. Bonetto et al. (2015) and Bonetto et al. (2017) presented semi-automatic approaches that extract lineaments from DTMs utilizing the curvature of geological features. Hashim et al. (2013) presented an edge detection and line linking method using Enhanced Thematic Mapping (ETM).

Colorimetry of an image can be used to detect features. By partitioning features in the image, e.g., matrix rock as lighter shades of gray and fractures as darker shades of gray, fracture pixels may be extracted separately from matrix rock using pixel values. Vasuki et al. (2017) developed an interactive colour based image segmentation tool using superpixels (Ren and Malik,

2003) which are groupings of pixels that are perceptually similar.

Edge detection techniques identify points in images where sharp changes in image intensity occur. Some of commonly used edge detection techniques in image processing are Canny, Sobel, Prewitt, Robert, Kuwahara, and Laplacian of Gaussian filters. Alternatively, edges may be detected using methods that are invariant to contrast and illumination in images. Phase symmetry and phase congruency algorithms (Kovesi, 1999, 2000) fall under this category. Phase symmetry is an edge detection technique that is invariant to local signal strength. The method works identifies the axis of a feature by isolating pixels symmetric along profiles that are sampled from all orientations except parallel to the feature. The axes of symmetry are regions where frequency components either approach a maximum or minimum. The phase congruency method is another edge detection method that detects features by identifying points where Fourier components are maximally in phase. This approach is also invariant to the magnitude of the signal. The property of invariance enables the identification of structures within the image even in the presence of noise. Vasuki et al. (2014) utilized an edge detection algorithm using the phase congruency principle coupled with a multi-stage linking algorithm for detection of fault maps.

The Hough transform (Duda and Hart, 1972) is another technique that has been used to detect lineaments in images. The Hough transform identifies pixels in binary images that are likely to represent rock fractures using a voting procedure. Each pixel in a binary image is represented as a sinusoidal curve in a 2D parametric space (or a Hough space). The voting procedure accumulates a vote for each curve in the parametric space corresponding to each non-zero pixel in the binary image. The curves with the highest votes are selected as probable fractures since they correspond to the largest number of non-zero pixels. Results by Callatay (2016) using the Hough transform for fracture detection report the following limitations. Firstly, the detection is limited to a given fracture orientation set owing to the definition of the Hough transform parameter space. Secondly, the issues of false positive detection and discontinuities persisted. The method is also limited by the fact that it needs a binarized image to start.

The development of wavelet theory in the field of harmonic analysis have led to applications in edge detection (Daubechies, 1992; Heil et al., 2006). Mallat and Hwang (1992) proposed wavelet-based approaches for edge detection. Wavelet-based methods differ from gradient-based edge detection methods that searches for local maxima of the absolute value of the gradient. Felsberg and Sommer (2001) introduced monogenic wavelets for the purpose. Tu et al. (2005) considered the use of magnitude response of complex wavelet transforms. Wavelets, owing to their isotropic properties, cannot extract curve-like features due to the lack of directional information (Labate et al., 2005). A number of wavelet-based approaches that have been proposed to overcome this lack of directional information such as curvelets (Candès and Donoho, 2005), ridgelets (Candès and Guo, 2002), contourlets (Do and Vetterli, 2005), bandlets (Le Pennec and Mallat, 2005) , wedgelets (Donoho, 1999), shearlets (Guo et al., 2005), and band-limited shearlets (Yi et al., 2009).

## 2.2  The complex shearlet transform

In images of fractured outcrops, the presence of discontinuous gaps due to rupture within the rock mass, which occur naturally and which maybe enlarged through weathering processes, are commonly used as defining criteria by interpreters to digitally trace and classify as fractures within the rock mass. Fractures may also be partially or completely sealed by the presence of infilling material that maybe mineralogically different from the adjacent rock material. In such a case, the contrast in colour and texture of the infill material provides an interpretative criterion for classification of these material regions as fractures. The presence of such prominent discontinuities within otherwise smooth regions of rock images, can be exploited by the complex shearlet transform to precisely identify position in the form of edges and ridges.

The basis of the complex shearlet transform applied to fracture extraction from images emanates from wavelet theory. Wavelets are rapidly decaying wavelike oscillations possessing a finite duration. Wavelet transforms are routinely used in digital signal processing applications which are often time-domain signals. They can also be applied to image data which can be considered as 2D functions. Wavelet transforms are not able to detect directionality of structural features in image data since they may only be dilated or translated. Shearlets that were introduced by Labate et al. (2005) as a new class of multidimensional representation systems, overcame a major shortcoming of wavelets by enabling dilation, shear transformation, and translation operations. The isotropic dilation of wavelets was replaced with anisotropic dilation and shearing in the case of shearlets. These modifications have resulted in shearlets possessing a number of properties that make them better suited to handle sparse, geometric features in 2D image data compared to traditional wavelets (Kutyniok and Labate, 2012).

The complex shearlet transform is a complex-valued generalization of the shearlet transform that was developed by Labate et al. (2005) to handle geometric structures in 2D data. Reisenhofer (2014) and King et al. (2015) proposed the idea of creating complex shearlets by modifying the shearlet construction so that real parts of the generating function are even-symmetric and imaginary parts of the generating function is odd-symmetric. They used the Hilbert transform to convert even-symmetric functions into odd-symmetric and vice versa. The complex shearlet measure for ridge and edge detection implemented in Reisenhofer (2014); King et al. (2015) and Reisenhofer et al. (2016) merged the ideas of phase congruency (Kovesi, 1999) and complex shearlets.

The complex shearlet measure first introduced by Reisenhofer (2014) and improved by King et al. (2015) was used for applications like coastline detection King et al. (2015), flame front detection Reisenhofer et al. (2016), and feature extraction from terrestrial LIDAR inside tunnels Bolkas et al. (2018). Karbalaali et al. (2018) used the complex shearlet transform for channel edge detection from synthetic and real seismic slices. Reisenhofer et al. (2016) presented a comprehensive comparison of complex shearlet based feature detection compared with conventional edge detectors such as Canny (Canny, 1986), Sobel (Sobel and Feldman, 1973), phase congruency (Kovesi, 1999), and another shearlet based edge detector (Yi et al., 2009). Bolkas et al. (2018) also made specific comparisons between the performance of Canny, Sobel, Prewitt (Prewitt, 1970) edge

detection methods versus space-frequency transform methods such as wavelets, contourlets, and shearlets. A detailed overview of the complex shearlet transform is provided in Appendix.A for the interested reader.

## 3   Methods

### 3.1   The automatic detection process

The automated fracture trace detection method that we present has five main steps (see Fig. 1). The first step of the method uses the Complex Shearlet-Based Ridge and Edge Measure (CoShREM), a MATLAB implementation by Reisenhofer et al. (2016) that utilizes functions from Shearlab3D developed by Kutyniok et al. (2016) and Yet Another Wavelet Toolbox (Jacques et al., 2011). The first step, namely the ridge detection, is dependent on a number of input parameters tabulated in Table 1 and Table 2. Equation (A28) gives the expression for the ridge measure.

An optimal set of deterministic parameter values which can extract features on all scales is not known *a priori*. Therefore, we vary the input parameters corresponding to the construction of the shearlet system and the ridge detection parameters within user-defined ranges to compute multiple ridge realizations. A ridge ensemble map is obtained by superposing the ridge images and normalizing. A simple sigmoid function is applied on the normalized ridge ensemble to non-linearly scale and thereby isolate higher image intensities. A user-defined threshold is then applied to the intensity values of this non-linearly scaled, normalized ridge ensemble image to extract a highly probable, binarized, ridge network. The threshold is set by a visual comparison of the input image with the extracted ridges. The range for each parameter in Table 1 and Table 2 is ascertained by first testing the effect of variation of each parameter with respect to a chosen base case image. This approach to automated detection captures features of multiple scales and highlights regions of uncertain feature extraction within the image.

The second step is the segmentation of the detected ridges using Otsu thresholding (Otsu, 1979). This operation removes small, disconnected, and isolated ridge pixel clusters. The third step is a skeletonization procedure where clusters of pixels representing the segmented ridges are thinned into single pixel representations. For intersecting fractures, the skeletonization procedure preserves the topology of the fracture network by recognizing and splitting the frame at the branch point. This step ensures that in subsequent DFN representation, there is no further effort expended in manually connecting the detected segments.

The fourth step involves piecewise linear polyline fitting to the skeletonized clusters. By default, our code attempts to fit polylines rather than lines to the pixel clusters. Polyline fitting retains geologically realistic, veering and curvature of fractures in the vectorized result. We use functions from the Geom2D toolbox (Legland, 2019) for the skeletonizing and polyline fitting. The fifth step is a line simplification procedure applied to the piecewise linear polyline clusters. A large number of polyline points would increase the size of vectorized files; hence, we use the Douglas-Peucker line simplification algorithm (Douglas

and Peucker, 1973) implemented by Ahmadzadeh (2017). The algorithm simplifies a piecewise linear polyline into one which has fewer segments. The number of polyline points assigned to each skeletonized cluster is set constant in the code, but this may be modified to be a linear function of the cluster size measured in pixels. If the image is georeferenced or the image scale is known, the code georeferences the simplified polylines and writes to a vectorized shapefile format.

The DFN in the vectorized shapefile format may now be used for any application that requires explicit fracture network geometry. An example of a fractured Posidonia shale micro CT (computed tomography) image slice from Dwarkasing (2016) (see Fig. 2) illustrates the effects of each of the steps involved.

## 3.2 Sensitivity analysis of parameters on extraction results

Since the detection results may vary owing to different parameter combinations, we conducted a sensitivity analysis to investigate the ridge extraction output with variation in parameter input. An example of a fractured image sample representing Mesoproterozoic sandstone from the Tomkinson Province, Northern Territory, Australia (Fig.3a) is chosen to study the effect of shearlet parameter variation. The image dimensions are 1313 x 1311 pixels and has four prominent fractures with two of them forming an intersection. A subtler fracture is present towards the top-left and a thick fracture located at the bottom-left of the image. A base case set of parameters for constructing a shearlet system and for ridge identification is set up in the table adjoining Fig.3a. We vary all parameters one by one with respect to this base case. Ridge extraction using the base case shearlet system shows that the major intersecting fracture system is identified; however, the largest fracture is detected only partially and that too, only at the peripheries. (see Fig.3b). The subtle fracture is detected but disconnected. A large amount of noise is also present.

The complex shearlet system is constructed by the tensorial product of a Mexican hat wavelet and a gaussian filter. The first two parameters *waveletEffSupp* and *gaussianEffSup* refer to the pixel widths over which the wavelet amplitudes sharply change from zero. The even- and odd-symmetric elements of constructed shearlet system using the base case parameters for the siliclastic example is depicted in Fig.4(i) and Fig.4(ii). We chose to maintain a ratio of two between *waveletEffSupp* and *gaussianEffSup*. The effect of increasing the effective support on the complex shearlet system is shown in Fig.4(xvii) - Fig.4(xix). Fig.4(xx) and Fig.4(xxi) indicate the effects of large ratios between the wavelet effective support and gaussian support. The second parameter is the *scalesperOctave* which determines the number of intermediate scales per octave. An octave is the interval between two frequency peaks. For example, we may consider a wavelet that is scaled by a factor of 2. Physically, this means a stretching of the wavelet thereby decreasing the frequency. The base-2 logarithmic ratio of the reduced frequency with respect to the original frequency, is the number of octaves by which the frequency has reduced. We set the number of octaves as a constant value of 3.5. This implies that there are 7 scales for the complex shearlet system as can be seen in Fig.4(iii) - Fig.4(ix). The *shearLevel* parameter indicate the discrete number of orientations that the complex shearlet system can assume. The selected value of 3 indicates that there are $2^3 + 2$ (ten) orientations possible for the complex shearlet system (see Fig.4(x)

- Fig.4(xvi)) and $2(2^3 + 2)$ (or 20 shearlets). For large images and large number of shearlets, computational effort is quite expensive. The *alpha* parameter is the degree of anisotropy induced by scaling with a null value of *alpha* maximizing the degree of anisotropy. We vary *alpha*, *shearLevel*, and the *scalesperOctave* but the effects on the constructed complex shearlet system are minimal as can be seen from Fig.4(xxii) - Fig.4(xxx).

The effects of variation of the parameters on ridge extraction is depicted in Fig.3(c) - Fig.3(p). Decreasing the value of the support by half identifies finer features, but then the largest fracture is completely missed (Fig.3e). When the support is doubled, the emphasis on larger features is more pronounced (Fig.3f). The effects of increasing and decreasing *scalesperOctave* is depicted in Fig.3g and Fig.3h with a higher value resulting in a finer ridge map. The effect of increase and decrease in the
number of shear levels on the final ridge map is quite minimal as can be seen from Fig.3i and Fig.3j. The effect of aniostropy parameter *alpha* is depicted in Fig.3k and Fig.3l with minimal anisotropy resulting in a finer ridge map. The *minContrast* parameter is a grayscale threshold (values from 0 - 255) applied to Eq.A28 to extract ridges. A larger value suppresses noisy features as can be seen from the comparison between Fig.3m and Fig.3n. The *offset* parameter is a scaling offset between odd-symmetric and even-symmetric shearlets quantified in octaves. Reducing the value of this parameter results in a coarser
ridge map with enhanced connectivity (Fig.3o) compared to the larger value which results in a finer map (Fig.3p).

From an interpreter's point of view, three different scales of fracturing need to be identified and false features also need to be suppressed. From the sensitivity analysis, the parameters that are most important to generate high-probability ridge maps, are the wavelet supports (required to capture multiple scales of fracture), grayscale contrast (suppressing noise and thereby
false features), and even-odd offset (which suppresses ridge detachments). This example illustrates the necessity of computing a ridge ensemble instead of searching for an ideal parameter combination.

### 3.3   Shearlet parameter selection

To decide upon the shearlet parameter space to generate multiple ridge realizations, we chose one sample image (see Fig. 5a). Base case parameters are chosen based on recommendations underlined in Reisenhofer et al. (2016) for shearlet construction
and ridge detection and these are tabulated in Table 3. The use of these results in the overlay depicted in Fig. 5b. As can be observed from visual inspection of the overlay of the detected ridges over the original image, the automatic method can extract a large number of fractures. However, there are still some false positives (features detected on the trees and inside the large karstic cavities) and false negatives (undetected small scale fractures).

To select the parameter ranges, we vary parameters with respect to the base case ridge image, thereby generating multiple ridge images. We use the structural similarity measure (Wang et al., 2004) to quantify the difference between the base case ridge image and other ridge images. Structural similarity (SSIM) is a measure commonly used in image quality assessment that returns one value as a measure of similarity between two images, where one image is the reference image. The SSIM is calculated for each ridge realization image corresponding to each parameter with respect to the base case ridge image. The

SSIM for variation in scaling offset, anisotropy scaling $\alpha$, Mexican hat wavelet support, gaussian filter support scales, minimum contrast, scales per octave, and number of shear levels are depicted in Fig. A3 according to the range of parameters in Table 4. From the analysis of the effects of parameters, we decided to vary the shearlet construction parameters so that we have 70 complex shearlet systems (see Table A1 for the parameters used to construct the 70 complex shearlet systems).

The total number of stochastic runs for the ridge detection is the number of combinations of shearlet systems and ridge specification parameters. Using such an approach, a probability map of detected features may be obtained based on which cut-off thresholds can be defined to remove false positives. The result of such a stochastic run with 1050 realizations is depicted in Fig. 5. From this result, the utility of the method is evident wherein the features that are obscured by shadows and the shrubbery

has a low strength signal which can then be filtered away thus reducing the number of false positives. Another advantage is that both large scale and fine features are captured which may not be possible using a single set of shearlet parameters.

## 4 Results

### 4.1 Trace extraction results from Parmelan, France

#### 4.1.1 Geological setting of the Parmelan plateau

We tested the automated fracture extraction method on an example from a carbonate outcrop from the Parmelan plateau in the Bornes Massif, France. The Bornes Massif is a northern subalpine chain in the western French Alps. The method was applied on a photogrammetric orthomosaic derived from a 3D outcrop model. The outcrop model was built from source photos acquired using a DJI Phantom 4 UAV. The image resolution is 18.6 mm/pixel. Processing of the drone images and generating the orthomosaic was done using AgiSoft PhotoScan Professional (Version 1.2.6) (2016*) software. The Parmelan Anticline in

France (see Fig. 6) is situated in the frontal part of the Bornes Massif and consists of Upper Jurassic to Cretaceous rocks of the European passive margin (Huggenberger and Wildi, 1991; Gidon, 1996, 1998; Berio et al., 2018).

This NE – SW trending anticline consists of a wide, flat crestal plateau bounded by steeply dipping limbs. Carbonates form the roof of a kilometre- scale box fold formed during the Alpine orogeny (Bellahsen et al., 2014). On the crestal plateau, a 1.7

25 km by 2.3 km large pavement of flat-lying shallow-water carbonates is exceptionally well exposed. The Parmelan outcrop is a good example of fracture patterns formed in a fold-and-thrust setting. We applied the automatic fracture detection technique on an orthomosaic that has been stitched together from drone photogrammetry over six different drone missions over the Parmelan. The combined extent of the six orthomosaics is depicted in Fig. 7a, and the areal extent of each orthomosaic is depicted in Fig. 7b.

### 4.1.2  Automatic extraction results on the Parmelan orthomosaic

Considering memory requirements and for faster computation, the image domain was divided into georeferenced sub-tiles using the Grid Splitter plugin in QGIS software. Visual filtering was carried out to remove tiles that did not have exposed rock, had a large degree of shrubbery, and which were at the orthomosaic edges where image resolution is poor. A total of 1000 tiles were chosen for the automated interpretation process. The areal extent of the orthomosaic covered 0.589 km$^2$, and this region is depicted in Fig. 7. The region covered by the tiles amounts to 0.379 km$^2$ and this is shown as an overlay of the selected tiles in Fig. 8a. Structural measurements were recorded at four small scale stations (around 2-5 sq.m per station) depicted in Fig. 8c-f.

An ensemble of 1050 ridges was computed using a set of shearlet parameters. A threshold for the ridge intensity was chosen to filter out the false positives. The threshold was determined by a visual examination of the overlay of detected ridges over the original images. The subsequent post-processing steps yielded features in each tile. These were geo-referenced and stitched back into a single vectorized file representation. Around 3 million features were extracted from the Parmelan orthomosaic. The $P_{21}$ fracture intensity was computed using the box-counting method by dividing the tile into a 25 x 25 (pixels) regular grid. The $P_{21}$ fracture intensity plot highlights the spatial variation of fracturing over the Parmelan plateau (see Fig. 8b). The vectorized fracture shape files along with the Parmelan basemap are presented as a public dataset (see Prabhakaran et al., 2019a).

### 4.1.3  Comparison with manual Interpretation and structural observations

To compare results of the automated approach to a manual interpretation, we chose a sub-region within the Parmelan ortho-mosaic. The selected subregion depicted in Fig. 9a consists of a 24 m x 24 m tile of the Parmelan orthomosaic. The image indicates fractures that seem to be isolated, without a well-connected topology, and which are predominantly aligned along an NW-SE direction. The fracturing intensity is variable across the tile. The contrast between fractures and the host rock fabric is intensified by the karstification of the fractures, which can be attributed to weathering and dissolution. Fig. 9b depicts an overlay of the automatically interpreted fractures overlain over the original tile. A total of 2910 features was extracted in this tile. This example highlights some of the technical challenges associated with automated fracture trace detection. Shrubbery is present in the image which obscures certain relevant features. The north-western corner of the image is blurred since it forms the extent of the orthomosaic.

The image also depicts open cavities or blobs, which could be the result of localized weathering. The effect of the cavities on the feature extraction is that only an edge is detected. Overall the fracture extraction efficiency is quite dependent on the resolution and quality of images. In the case of the Parmelan data acquisition, the UAV was flown at an altitude of 50-70 metres above the pavement; therefore, features such as closed veins, and slightly open fractures are below the resolution of the drone camera. A higher image resolution is necessary to extract such features. In our specific case study, good lighting and exposure during the UAV flight mission prevented shadows from obscuring the imagery. Fig. 9c depicts a manually performed interpretation at a zoom level of 1:2000 on the raster image with a total of 341 features. $P_{21}$ fracture intensity comparisons

of both automatic and manual traces are shown in Fig. 9d and Fig. 9e. The difference between the automatic and manual interpretation highlights the inclination of the interpreter to neglect small scale features. Based on geological experience and prior knowledge of the field area, there is a tendency to interpret and link together disconnected features from the original raster image. The closest small-scale station to the sub-tile depicted in Fig. 9a is station 2. There is agreement between the rose plots

of station 2 (see Fig. 8c) and the interpretations (Fig. 9e and Fig. 9f). The observed fractures in both cases are predominantly sub-vertical.

### 4.1.4    Application to mineralized fractures

We now showcase an example of a close-range image containing mineralized veins that are invisible to photogrammetry at altitudes of 40 - 70 m . The resolution of this image is 0.18 mm/pixel and was taken using a handheld DSLR camera.

In this high-resolution image, the fracture infill has similar colour as the host rock as can be seen in Fig. 10a. A manual interpretation of the veins (at a zoom of 1:750) is depicted in Fig. 10b. Using a well-tuned set of parameters with reduced wavelet effective supports, it is possible to extract the much thinner and subtle features as depicted in Fig. 10c. It can be observed from comparison between Fig. 10b and Fig. 10c, that a large number of false features are also highlighted alongside the features of interest. The main contributors to the extraction of these non-fracture features are the natural rugosity of the

rock face, presence of pebbles, pockmarks, and erosion features. The arrangement of these artefacts display a very different pattern; small lines with random direction compared to the fractures which are consistently oriented and quite continuous. The veins are also of different thicknesses, with a few veins anastomosing and some branching in a horsetail manner. Some of the thicker veins also exhibit microstructure within the mineral infill. Further tuning of parameters in order to capture all the veins while also suppressing false features is quite challenging and hence we do not explore this in further detail. Despite the noise,

the automated method is not limited to capturing only open fractures but can also extract closed fractures.

### 4.2    Trace extraction results from Brejões, Brazil

### 4.2.1    Geological setting of the Brejões pavement

The second case study for the automated extraction method is a carbonate outcrop from the Irecê Basin, Central Bahia, Brazil (see Fig. 11a, Fig. 11b). The Irecê Basin is located within the northern region of the São Francisco Craton. The Brejões

pavement study area is within the Irecê Basin and consists of Neoproterozoic platform carbonates of the Salitre Formation (750-650 Ma). The Neoproterozoic cover was affected by the Brasiliano Orogeny (750-540 Ma) in two separate folding events resulting in fold belts around edges of the São Francisco Craton (Ennes-Silva et al., 2016). The Brejões pavement UAV imagery that we used for our analysis was acquired by Boersma et al. (2019). Structural measurements from Boersma et al. (2019) is shown in Fig. 11c. The orthomosaic covers an area of 0.81 km$^2$ and consists of fractured, black oolitic limestones that

correspond to Unit A1 of the Salitre stratigraphy (Guimarães et al., 2011). The resolution of the Brejões orthomosaic is 20.3 mm/pixel.

### 4.2.2 Automatic extraction results on the Brejões orthomosaic

The Brejões orthomosaic is split into 222 tiles for the analysis and this region is shown in Fig. 11d. The Brejões example has a different fracturing style than the Parmelan and consists of an intricate pattern of multi-scale conjugate fractures. The shearlet combinations utilized in the case of the Parmelan was insufficient to capture this variation in scales. Specifically, in the Brejões case, the large scale features were not captured. A visual inspection of the ridges was necessary to identify the shearlet combinations that amplified the large scale features. The contribution of these ridges was increased (factor of 8) in the ridge ensemble to highlight these large deformation features. Fig. 11e depicts the $P_{21}$ fracturing intensity computed using the box-counting method by dividing each tile into a 25 x 25 (pixel) regular grid. The vectorized fracture shape files along with the Brejões basemap are presented as a public dataset (see Prabhakaran et al., 2019b).

### 4.2.3 Comparison with manual interpretation and structural observations

The automatically extracted features from the Brejões image data was compared with manual interpretations performed by and obtained from Boersma et al. (2019) at seven stations. The automatic interpretations were trimmed to the peripheries of the manual interpretations for a fair comparison between both vectorizations. The location of these stations alongside the automatic versus manual interpretations are shown in Fig.12. A comparison of the rose plots and cumulative length distributions of the manual and automatic interpretations is depicted in Fig.13. A few observations can be made from the comparison. Firstly, similar to the Parmelan case, the interpreter picks a lesser number of features. Secondly, there is a tendency to extend fractures across image regions where there is no real evidence of rock failure. Thirdly, there is an inconsistency in specifying the connecting topologies between the interpreted traces.

In some stations (see Mid #2, Mid #3 and North in Fig.12), the automated interpretation suffers from a large number of false positives. A close examination indicates that the presence of shadows and eroded, undulating topography of the rocks are the main reasons for these false positives. In the Brejões case, the drone was flown at around 10.00 AM, and hence the exposure of the outcrop face was not optimal. The inclined illumination enhances shadows on the rugged topography, which are then seen as false positives in the automatic interpretation. False positives due to shrubbery are minimal in the station regions considered.

### 4.3 Benchmarking with data from Thiele et al. (2017)

We further tested the automated trace detection on a recently published case study from Thiele et al. (2017). The images selected are orthophotographs of two 10 x 10 m areas from Bingie Bingie Point, New South Wales, Australia (see Fig. 14a and Fig. 15a). The exposed rocks are Cretaceous to Paleogene dykes, intruding diorites, and tonalities cross-cut by joint sets (Thiele et al. 2017). The images are complex as they contain both open and closed fractures of different scales, distributed between multiple lithological layers. The images also contain water, shadows, and debris, which makes it even more challenging. We chose this dataset to benchmark the quality of our results with those presented using the semi-automatic cost function based

trace mapping approach of Thiele et al. (2017).

The variation in fracture scales implied that similar to Brejões, a different set of shearlet combinations were needed. We generated 2700 ridge realizations which were used to construct a normalized ridge ensemble map for both images (see Fig. 14b and Fig. 15b). A simple, non-linear sigmoid function was applied to the normalized ridge intensity to enhance ridge strength (see Fig. 14c and Fig. 15c) and a threshold was chosen based on visual comparison with the source image to yield highly probable, binarized ridge images (see Fig. 14d and Fig. 15.d). The subsequent workflow steps, as described in Sect. 3.1 were followed to obtain vectorized traces (see Fig. 14e and Fig. 15e). The vectorized traces were used to render assisted interpretations depicted in Fig. 14f and Fig. 15f which are comparable in quality to the assisted interpretation of Thiele et al. (2017).

In the published results of Thiele et al. (2017), assisted interpretations of both areas are achieved in 37 minutes and 34 minutes, respectively. We can report better performances of 27 and 32 minutes for the same areas. The time does not include computing of the ridge realizations. Once the high probability trace map was generated, the subsequent steps of the automated detection workflow took around 3 minutes. The remainder of the time was used to perfect the assisted interpretation. The post-processing tasks performed in this second step were the removal of false positives owing to shadows, water, and debris and joining of segments which were disjointed due to poor resolution within the image. Though we have performed a benchmarking exercise with the data from Thiele et al. (2017) and also compared our results with manual interpretation, it would be useful to compare with more manual interpretations to further validate the accuracy of the technique. Such comparison, however, can be done only on networks which are either limited in their spatial extent or in the number of features interpreted. For large orthomosaics, a benchmarking exercise can be challenging as few manually rendered datasets are comparable in network size.

## 5  Discussion

Extraction of fracture traces from photogrammetric data is a necessary processing step to construct DFN representations. DFNs created using fracture patterns that are directly extracted from rock images, are advantageous as they honour the spatial architecture of fracture networks. Automated extraction methods reduce the human component in data processing, and we have achieved this using the complex shearlet transform ridge detection method accompanied by post-processing steps. The complex shearlet method can detect both edges as well as ridges in fractured rock images. We find that the ridge measure works very well for extraction of fractures, and we use the ridge measure in all our case studies. Though the method performs very well and can extract much more traces than is possible manually while reducing interpreter bias, there are some issues that need to be mentioned. In this section, we discuss on the validity and limitations of the technique, areas where there is scope for further development, and also describe some potential extended applications of the method.

## 5.1 Validity and limitations

– **Detection of mineralized features** The method works well when the features of interest are barren and a prominent. When fractures are closed and filled then they are generally harder to detect and require high resolution images (< 1 mm/pixel) which can be recorded only at very close ranges at very-low UAV flight altitudes. Recent outcrop studies (Ukar et al., 2019) indicate that many of the barren features in outcrop are absent within the same subsurface lithological unit while maintaining good correspondence between mineralized features in both outcrop and subsurface. When mineral fill has a marked colour contrast with respect to the host rock (as in vein data published recently by Meng et al., 2019), then superpixel segmentation algorithms can be successful (Vasuki et al., 2017). In the case of poor contrast, the complex shearlet transform would require a great deal of manual tuning of detection parameters to extract reliable results. At such close ranges, as is needed for veins extraction, it is also likely that many more noisy features un-related to fracturing would arise.

Since mineral-fill of fractures can provide a clearer picture into evolution, timing, and stress history of fractures, identifying them on an outcrop scale is important. This is doubly significant, when the goal is to directly extrapolate fracture patterns from a particular outcropping to the same subsurface target. In such a case, close range UAV-mounted hyperspectral data acquisition would be better suited than conventional imaging and image processing methods. With hyperspectral imaging, data is collected in near-continuous spectral bands. The spectral response of minerals constituting the rock, owes to atomic-molecular level processes triggered on interaction with a light source (active or passive) and this may be utilized to identify mineral composition. Since mineral fill of veins are likely to have a different spectral response from the mineralogy of the host rock, this variation may be used to isolate the pixels that correspond to veins.

A recent review on close range hyperspectral imaging for mineral identification identifies various previous studies performed for specific minerals (Krupnik and Khan, 2019). It would be interesting to observe, identify, and distinguish between mineralized sequences based on the differences in spectral response of the fracture infill material. Since hyperspectral data is much more voluminous and with significantly more complex image processing than conventional photogrammetry, such analysis could be confined to selected regions within the outcrop. In conjunction with conventional UAV photogrammetry that covers larger spatial area, laboratory based geochemical studies, and outcrop observations (scanline sampling, abutting relations etc.), a more detailed fracture characterization may be conducted.

– **Detection of large cavities and false features** Both the Parmelan and Brejões pavements exhibit karstification with the Parmelan containing many more collapsed karstic regions. The presence of such low-aspect ratio discontinuities are quite rare in siliciclastic and volcanic outcrops but can prove problematic to the application of the method in carbonate outcrops where karstification is severe. Both the ridge and edge measures would fail in identifying such blobs or would at best, extract the periphery of the cavity. In recent work by Reisenhofer and King (2019), blob detection measures have been developed within the shearlet framework and could potentially solve this issue.

Another issue is the effect of undulating topography and shrubbery in generating false positives. False positives generally appear when there is shrubbery, shadows, very rugged terrain, and non-fracture bedding planes. In the case of the Parmelan, the use of multiple ridges was successful in suppressing the false positives owing to shrubbery. However, in Brejões, false positives due to underbrush were more difficult to suppress because they shared the same scale as that of the fractures. In Brejões, shrubbery was also present within some of the wider fractures causing false negatives. In such cases, manual interference is necessary to either mask the regions of shrubbery before the automatic extraction or to remove (or connect) the vectorized traces after the automated extraction. Additionally, carbonate outcrops are prone to widespread erosion owing to exposure to meteoric water from precipitation cycles and air corrosion. Geomorphological features owing to these erosive processes may also play a role in generation of false positives.

– **Parameter selection**

A significant difference in fracture scales within an image of interest can prove problematic for the method. In such a case, a vast number of ridge detection runs and associated increase in computational time is needed to construct a ridge ensemble that takes into account all scales of discontinuities and yields a satisfactory result. When such variation is localized, the image could be segmented into regions that correspond to varying fracture intensities and processed separately. This may be difficult to assess *a priori* and in such cases, would require trial runs. In the Brejões outcrop example and the close range Parmelan vein example, this difference in fracture scales was ubiquitous throughout the exposure and more pronounced than the Parmelan outcrop. Using visual comparison with the original image, the effect of ridges resulting from certain shearlet parameter combinations was enhanced, so that the ridge ensemble is improved. In Brejões, it was the large scale features that needed to be strengthened while in the case of the Parmelan vein example, the smaller features needed sharpening. Since parameter selection is still done manually, a more comprehensive way of arriving at the optimal shearlet combination is desirable. An algorithm that automatically optimizes for shearlet parameters corresponding to each individual scale of fracture is worthy of attention.

– **Artificial fragmentation of traces**

Manual fracture interpretation from images often involve the step of classifying fracture traces into separate sets based on ground truth observations or with respect to fracture strike. The automated method described here in its current form can only extract traces and cannot distinguish / classify traces as belonging to separate sets. When fractures intersect each other, the issue of artificial fragmentation of seemingly continuous traces arises. If an image consists of two orthogonally intersecting fractures, the automated method would result in four traces intersecting at a single branch point, even though a manual interpretation would only identify two fracture traces belonging to two different geometric sets. This type of fragmentation would result in different length distributions as observed in (insert figures that compare automatic and manual interpretations). This kind of fragmentation is not an issue if such an outcrop DFN is used for geometric input for flow / geomechanics simulation. This is because the process of meshing models with explicitly specified DFN geometry would, in any case, require the specification of all intersection points (or forced fragmentation of long intersecting

fractures). Therefore, the practioner must exercise caution when using cumulative length distributions derived from outcrop DFNs that are automatically extracted.

A single fracture could also be fragmented without being cut by other intersecting fractures. This may happen in the case of false negatives (due to shadows falling over part of fracture, debris or shrubbery within an open fracture, and when fracture opening is very thin at some regions along fracture length) that cause fragmentation of fractures with gaps in between them. This kind of fragmentation affects the topology of the network in addition to depressing the height cumulative length distribution. It maybe noted that, specification of fracture endpoints manually is also fraught with bias (Peacock et al., 2019). A solution would be to use a range of linking thresholds to connect traces and study the effects of threshold values on network topology and length distribution.

## 5.2 Recommendations for future work

– **Link between extractable $P_{21}$, drone flying altitude, and camera resolution**

From the $P_{21}$ analysis on the Parmelan and the Brejões automatically extracted fractures, the maximum value $P_{21}$ was around eight m$^{-1}$. The same drone model was used in both cases (DJI Phantom 4), and the flying altitude was also similar (between 40 and 70 metres). Although such a conjecture needs further verification, there could be a relation between the resolution of imagery and maximum extractable fracture intensity. Often flight altitudes are chosen by drone pilots depending upon considerations such as local topography, weather conditions, and presence of impediments (such as trees, electricity poles, and telecommunications towers). A detailed analysis of the relation between flying altitude (and consequently image resolution) and extracted fracture intensity could provide drone pilots with insights and guidelines for UAV-based outcrop analysis. The ideal flying resolution to identify features of interest may be ascertained by carrying out a series of acquisitions at a location where ground truth is known.

– **Generating data for fractured reservoir modelling workflows**

Fractured reservoir characterization workflows in the oil and gas industry have traditionally used stochastic techniques that attempt to extrapolate averaged fracture statistics (either from borehole imagery, core data, or outcrop analysis) to reservoir volumes. The use of Multiple Point Statistics (MPS) for fracture network generation was highlighted by Bruna et al. (2019) as an alternative approach to DFN modelling. MPS uses training images of realistic fracture networks to learn patterns and then generate non-stationary fractured reservoir models. Corrected for false positives and noise, the automated method can produce accurate, geologically realistic, and unbiased training images that can feed into the MPS workflow. Since our method can extract large scale fracture networks (millions of features from sub- square kilometre regions), it is also well suited to provide training data for deep learning workflows. Recently, the use of Generative Adversarial Networks (GANs) for geological modelling at the reservoir scale was proposed by Dupont et al. (2018), Zhang et al. (2019) as an alternative to conventional geostatistics, MPS, and object-based modelling. GANs form a subset of deep learning architectures that are used for generative modelling (Goodfellow et al., 2014). GANs that are trained on realistic data can then generate geologically realistic, non-stationary models.

– **Deep learning methods for trace extraction**

Deep learning methods have revolutioned computer vision applications. Various neural architectures have documented high degrees of accuracy in machine vision tasks such as overall image classification, identification and classification of objects within an image images, localization of objects, extraction of regions of interest (semantic segmentation), and extraction of regions corresponding to individual objects (instance segmentation). The problem of fracture trace extraction falls within the problem category of region extraction of individual objects and hence may be attempted using techniques such as mask Regional Convolutional Neural Networks (He et al., 2017). Deep learning methods, in general, require large amounts of labelled data to train. In the case of a mask RCNN, the library of training images must contain marked regions (or overlays) indicating pixels of interest that correspond to individual fractures. The automated method described in this manuscript can be used to rapidly generate a large number of overlay images that can be used as training data for mask RCNN architectures.

## 6 Conclusions

This paper presents a method to automatically detect and digitize fracture traces from images of rock fractures using the complex shearlet transform. The technique replaces the task of manually interpreting fractures, which is time-consuming, prone to interpreter bias, and which suffers from a lack of repeatability. The case studies that are presented highlight the utility of the complex shearlet based measure for automatically detecting fracture traces from 2D images. The automatic trace detection method combines the complex shearlet ridge measure with a series of post-processing steps that include image segmentation, skeletonization, polyline fitting, and polyline simplification. We tested the method at different scales of rock displacement, at outcrop scale ($\sim 10^2$ m) and station-scale ($< 10$ m), using two orthomosaics reconstructed from drone photogrammetry and two rock pavement images. We have considered carbonate and igneous rock lithologies in the case studies. Using the method, we have extracted millions of 2D features from outcrop-scale drone orthophotos. The processing time of the technique depends upon the intensity of fracturing and the complexity of the fracture networks contained within the image. The automatic trace extraction results are quantitatively compared with manually interpreted fractures on selected sub-samples of the image domain using fracture trace density metrics. The automated technique is capable of extracting a much larger number of features, with a marked reduction in bias. The method outlined in this paper greatly simplifies the process of generating deterministic, outcrop-based DFNs. The automatically extracted, fracture patterns can be used by structural geologists to link deformation features to tectonic history and by geomodellers in sub-surface NFR modelling.

*Code and data availability.*

. MATLAB code that was used to generate the results in this manuscript is available on Github

https://github.com/rahulprabhakaran/Automatic-Fracture-Detection-Code (see Prabhakaran 2019)

. Fracture and image data correponding to the Parmelan and Brejões outcrops are available at the 4TU Centre for Research Data repository (https://researchdata.4tu.nl/en/)

- Fracture Network Patterns from the Brejões Outcrop, Irecê Basin, Brazil (see Prabhakaran et al. 2019a)

- Fracture Network Patterns from the Parmelan Anticline, France (see Prabhakaran et al. 2019b)

### Appendix A:  Overview of the Complex Shearlet Transform

### A1   The Continuous Shearlet System

A shearlet generating function consists of an anisotropic scaling matrix and a shear matrix. Let the shearlet generating function be:

$$\psi \in L^2\left(\mathbb{R}^2\right) \tag{A1}$$

The admissibility criteria for the shearlet generating function is :

$$\int_{R^2} \frac{\left|\hat{\psi}(\xi_1\xi_2)\right|^2}{\xi_1^2} \, d\xi_2 d\xi_1 < \infty \tag{A2}$$

where $\hat{\psi}$ is the 2D fourier transform of $\psi$.

A shearlet satisfying Eq.A2 is an admissible shearlet or a continuous shearlet (Kutyniok and Labate, 2012). The admissibility condition implies that a reconstruction formula exists for the associated continuous shearlet transform. In order to achieve an optimally sparse approximation of an image that possesses anisotropic singularities, the analysing elements must consist of waveforms that range over several scales, orientations, and locations with the ability to become very elongated. To this end, a combination of a scaling operator to generate elements at different scales, an orthogonal operator to change orientations, and a translation operator to displace elements over the 2D plane, is used. The scaling matrix $A_a$ is defined as (Labate et al., 2005):

$$A_a = \begin{pmatrix} a & 0 \\ 0 & a^\alpha \end{pmatrix}, \ \alpha \in [0,1]$$

The value of $\alpha$ controls the degree of anisotropy. (For more information on the anisotropy scaling molecules or $\alpha$–molecules see Grohs et al. 2016.) The scaling matrix is parabolic when $\alpha = \frac{1}{2}$.

An orthogonal transformation to change the orientations of waveforms. Rotation operators are not preferred as they destroy the structure of the integer lattice $Z^2$ whenever the rotation angle is different from $0, \pm\frac{\pi}{2}, \pm\pi, \pm\frac{3\pi}{2}$. Changes in the structure of integer lattice is problematic when transitioning from continuum to digital setting. Hence, a shearing transformation is used where the anisotropic shearing transformation matrix $S_s$ are defined as:

$$S_s = \begin{pmatrix} 1 & s \\ 0 & 1 \end{pmatrix} \text{ where the parameters } a \in \mathbb{R}^+, s \in \mathbb{R}$$

The shearing matrix $S_s$ preserves the structure of the integer grid for any $s \in \mathbb{N}$. The shearing matrix parametrizes orientations using the variable $s$ associated with slopes rather than angles and leaves the integer lattice invariant, provided $s$ is an

integer. The difference between isotropic and anisotropic dilation with shearing is depicted in Fig. A1a and Fig. A1b).

A shearlet system is defined as (Kutyniok and Labate, 2012):

$$SH(\psi) = \left\{ \psi_{a,s,t} = a^{-3/4} \psi \left( A_a^{-1} S_s^{-1} (\cdot - t) \right) a \in \mathbb{R}^+, s \in \mathbb{R}, t \in \mathbb{R}^2 \right\} \tag{A3}$$

where $(\cdot - t)$ denotes the translation by a point $t$.

The corresponding shearlet transform for mapping a function $f \in L^2(\mathbb{R}^2)$ into coefficients, $SH_\psi f(a,s,t)$ specified by scaling $a$, shearing $s$ and translation $t$ is given by:

$$f \rightarrow SH_\psi f(a,s,t) = f, \psi_{a,s,t} \tag{A4}$$

## A2   Cone Adapted Continuous Shearlet Systems

Equation A4 renders horizontal shearlets elongated at very fine scales, which is problematic in digital implementations. Because the shearing operator can range over a non-bounded interval, directions are not treated uniformly. To overcome this drawback of shearing, the cone adapted shearlet system was introduced in which the frequency plane is split into a horizontal and vertical cone that restricts the shear parameter to bounded intervals (see Fig. A2 a). Dividing the frequency plane in such a manner ensures uniform treatment of directions (Guo et al., 2005; Kutyniok and Labate, 2012). A cone adapted shearlet system can be

tiled by further division of the frequency domain. Such a tiling configuration (see Fig.A2 b) ensures that all directions are treated "almost equally" (Kutyniok and Labate, 2012). There is still small, but controllable bias in the coordinate axes directions). The cone adapted shearlet systems can therefore be expressed as the union of a horizontal cone, a vertical cone, and a low-frequency centre component. The frequency plane is thus split into four horizontal and vertical cones with a low-frequency square region in the centre. The low-frequency region is given by the relation (Kutyniok and Labate, 2012):

$$\mathcal{R} = \{ (\xi_1, \xi_1) : |\xi_1|, |\xi_2| \leq 1 \} \tag{A5}$$

Inside each cone, the shearing variable $s$ is only allowed to vary over a finite range. This produces elements with uniformly distributed orientations. The union of the generating functions for the horizontal cones $\psi \in L^2(\mathbb{R}^2)$, vertical cones $\tilde{\psi} \in L^2(\mathbb{R}^2)$ and for the square low frequency region $\varphi \in L^2(\mathbb{R}^2)$ is expressed as (Kutyniok and Labate, 2012):

$$SH(\varphi, \psi, \tilde{\psi}) = \Phi(\varphi) \cup \Psi(\psi) \cup \tilde{\Psi}(\tilde{\psi}) \tag{A6}$$

where

$$\Phi(\varphi) = \left\{ \varphi_t = \varphi(\cdot - t) : t \in \mathbb{R}^2 \right\}; \tag{A7}$$

$$\tilde{\Psi}(\psi) = \left\{ \tilde{\psi}_{a,s,t} = a^{-\frac{3}{4}} \tilde{\psi} \left( \tilde{A}_a^{-1} S_s^{-1} (\cdot - t) \right) : a \in (0,1], |s| \leq 1 + a^{\frac{1}{2}}, t \in \mathbb{R}^2 \right\}; \tag{A8}$$

$$\tilde{\Psi}(\psi) = \left\{ \tilde{\psi}_{a,s,t} = a^{-\frac{3}{4}} \tilde{\psi}\left( \tilde{A}_a^{-1} S_s^{-1}(\cdot - t) \right): a \in (0,1], |s| \leq 1 + a^{\frac{1}{2}},\, t \in \mathbb{R}^2 \right\}. \tag{A9}$$

Scaling matrix for vertical cones, $\tilde{A}_a$ is expressed as:

$$\tilde{A}_a = \begin{pmatrix} a^\alpha & 0 \\ 0 & a \end{pmatrix} \tag{A10}$$

The cone adapted continuous shearlet transform is expressed as the mapping:

$$5 \quad f \to SH_{\varphi,\psi,\tilde{\psi}}\, f\left(t', (a,s,t), (\tilde{a},\tilde{s},\tilde{t})\right) = \left( f,\ \varphi_{t'}, f,\ \psi_{a,s,t}, f,\ \tilde{\psi}_{\tilde{a},\tilde{s},\tilde{t}} \right) \tag{A11}$$

## A3   The Discrete Cone Adapted Shearlet System

A discrete version of the cone adapted shearlet system may be defined with scaling parameter $j$, shearing parameter $k$, and translation parameter $m$ for a sampling factor of $c = (c_1, c_2) \in (\mathbb{R}_+)^2$. Similar to Eq.A6 this is a union of the generating functions for vertical, horizontal, and low frequency central region.

$$10 \quad SH(\varphi,\psi,\tilde{\psi};c) = \Phi(\varphi;c_1)\ \cup \Psi(\psi;c)\ \cup \tilde{\Psi}(\tilde{\psi};c) \tag{A12}$$

$$\Phi(\varphi;c_1) = \left\{ \varphi_m = \varphi(\cdot - c_1 m)\ : m \epsilon \mathbb{Z}^2 \right\}; \tag{A13}$$

$$\Psi(\psi;c) = \left\{ \psi_{j,k,m} = 2^{\frac{3}{4}j} \psi\left( S_k A_{2^j} \cdot - M_c m \right): j \geq 0, |k| \leq \left\lceil 2^{\frac{j}{2}} \right\rceil,\, m \in \mathbb{Z}^2 \right\}; \tag{A14}$$

$$\tilde{\Psi}(\tilde{\psi};c) = \left\{ \tilde{\psi}_{j,k,m} = 2^{\frac{3}{4}j} \tilde{\psi}\left( S_k{}^T \tilde{A}_{2^j} \cdot - \tilde{M}_c m \right): j \geq 0, |k| \leq \left\lceil 2^{\frac{j}{2}} \right\rceil,\, m \in \mathbb{Z}^2 \right\}; \tag{A15}$$

with $M_c = \begin{bmatrix} c_1 & 0 \\ 0 & c_2 \end{bmatrix}$ ; $\tilde{M}_c = \begin{bmatrix} c_2 & 0 \\ 0 & c_1 \end{bmatrix}$ ; ($M_c$ and $\tilde{M}_c$ are sampling matrices for horizontal and vertical cones)

$$20 \quad A_{2^j} = \begin{bmatrix} 2^j & 0 \\ 0 & 2^{j/2} \end{bmatrix} ;\ \tilde{A}_{2^j} = \begin{bmatrix} 2^{j/2} & 0 \\ 0 & 2^j \end{bmatrix} ;\ (A_{2^j}\text{ and }\tilde{A}_{2^j}\text{ are dyadic scaling matrices for horizontal and vertical cones})$$

$$S_k = \begin{bmatrix} 1 & k \\ 0 & 1 \end{bmatrix} \text{ (shearing matrix)}.$$

The discrete cone adapted shearlet transform associated with $\phi$, $\psi$ and $\tilde{\psi}$ is given by the mapping,

$$f \to SH_{\varphi,\psi,\tilde{\psi}}\, f\left(m', (j,k,m), (\tilde{j},\tilde{k},\tilde{m})\right) = \left( f,\ \varphi_{m'}, f,\ \psi_{j,k,m}, f,\ \tilde{\psi}_{\tilde{j},\tilde{k},\tilde{m}} \right). \tag{A16}$$

## A4 The Complex Discrete Cone Adapted Shearlet System

Taking the complex valued wavelet of a real valued even symmetric wavelet generator $\psi^{even} \in L^2(\mathbb{R}^2)$, using the Hilbert transform operator $(\mathcal{H})$, a complex valued shearlet generator is obtained (from Reisenhofer, 2014; King et al., 2015)

$$\psi^c = \psi^{even} + i\,\psi^{odd}. \tag{A17}$$

The complex valued function can be written in terms of a Hilbert transform pair of an even-symmetric real valued shearlet and an odd-symmetric real valued shearlet: (from Reisenhofer, 2014; King et al., 2015)

$$\psi^c = \psi^{even} + i\,\mathcal{H}\psi^{even}. \tag{A18}$$

The Hilbert transform operator is written as,

$$\mathcal{H}(f)(t) = \lim_{a\to\infty} \int_{-a}^{a} \frac{f(\tau)}{t-\tau} d\tau. \tag{A19}$$

The discrete cone adapted complex shearlet system is given as (Reisenhofer, 2014; King et al., 2015):

$$SH(\varphi, \psi, \tilde{\psi}; c) = \Phi(\varphi; c_1) \cup \Psi(\psi; c) \cup \tilde{\Psi}(\tilde{\psi}; c) \tag{A20}$$

and

$$SH^c(\varphi, \psi, \tilde{\psi}; c) = \Phi(\varphi; c_1) \cup \Psi^c(\psi; c) \cup \tilde{\Psi}^c(\tilde{\psi}; c) \tag{A21}$$

where,

$$\Phi(\varphi; c_1) = \left\{ \varphi_t = \varphi(\cdot - c_1 m) : m \in \mathbb{Z}^2 \right\}, \tag{A22}$$

$$\Psi^c(\psi; c) = \left\{ \psi^c_{j,k,m} = \psi^c_{j,k,m} + i\,(\mathcal{H}_{(1,0)} \mathrm{T}\,\psi)_{j,k,m} : j \geq 0, |k| \leq \lceil 2^{\frac{j}{2}} \rceil,\, m \in \mathbb{Z}^2 \right\}, \tag{A23}$$

$$\tilde{\Psi}^c\left(\tilde{\psi}; c\right) = \left\{ \tilde{\psi}^c_{j,k,m} = \tilde{\psi}^c_{j,k,m} + i\,(\mathcal{H}_{(0,1)} \mathrm{T}\,\tilde{\psi})_{j,k,m} : j \geq 0, |k| \leq \lceil 2^{\frac{j}{2}} \rceil,\, m \in \mathbb{Z}^2 \right\}. \tag{A24}$$

Correspondingly the discrete complex cone adapted shearlet transform is given by the mapping,

$$f \to SH^c_{\varphi,\psi,\tilde{\psi}}\, f\left(m', (j,k,m), (\tilde{j}, \tilde{k}, \tilde{m})\right) = \left( f,\, \varphi_{m'}, f,\, \psi^c_{j,k,m}, f,\, \tilde{\psi}^c_{\tilde{j},\tilde{k},\tilde{m}} \right). \tag{A25}$$

$$\tag{A26}$$

## A5 Edge and Ridge Detection using the Complex Shearlet Transform

The behavior of the coefficients of the even symmetric and odd symmetric shearlets can be used to detect edges and ridges. An edge measure for an image $f \in L^2(R^2)$, a location $x \in R^2$ and a shear parameter $s$ is given as (Reisenhofer, 2014; King et al., 2015),

$$E_\psi(f,x,s) = \frac{\left| \sum_{a\epsilon A} Im\left(f, \psi^c_{a,s,x}\right) \right| - \sum_{a\epsilon A} \left| Re\left(f, \psi^c_{a,s,x}\right) \right|}{|A| max_{a\epsilon A} \left| Im(f, \psi^c_{a,s,x}) \right| + \varepsilon}, \tag{A27}$$

where $A \subset \mathbb{R}^+$ is a set of scaling parameters, $\psi$ is a real valued symmetric shearlet and $\epsilon$ prevents division by zero. The complex shearlet based edge measure can give approximations of the tangential directions of an edge. A line measure or ridge

10   measure is obtained by interchanging the role of the even symmetric and odd symmetric shearlets (Reisenhofer, 2014; King et al., 2015),

$$L_\psi(f,x,s) = \frac{\left| \sum_{a\epsilon A} Re\left(f, \psi^c_{a,s,x}\right) \right| - \sum_{a\epsilon A} \left| Im\left(f, \psi^c_{a,s,x}\right) \right|}{|A| max_{a\epsilon A} \left| Re(f, \psi^c_{a,s,x}) \right| + \varepsilon}. \tag{A28}$$

15   Both the edge and ridge measures given above are inspired from the phase congruency measure of Kovesi (2000). The edge and ridge measures are almost contrast invariant.

*Author contributions.* RP developed model code, performed the automatic extraction on all case studies, participated in data acquisition during the Parmelan fieldwork, and wrote the manuscript. P-OB took the lead role in data acquisition of the Parmelan dataset, carried out processing of the drone imagery, assisted in generating artwork, and provided manual fracture interpretations. GB took part in the data acquisition at the Brejões outcrop and provided knowledge and expertise about the regional geology of the Irecê Basin. GB and DS provided

expertise and supervision concerning the development of the workflow and writing of the manuscript.

*Competing interests.* The authors would like to declare that there was no competing interests involved.

*Acknowledgements.* ENI Exploration is thanked for partial funding of the Ph.D. project of the first author, for the financing accorded to the fieldwork carried out in Parmelan, France and for permission to use the Parmelan photogrammetric data to test the automatic trace detection method. The authors would like to accord their special gratitude to Marco Meda (ENI Exploration) for his gracious hospitality and

coordination of the SEFRAC consortium meetings held at Milano and for the stimulating discussions. Thanks are also due to Silvia Mittempergher (University of Milano Bicocca), Fabrizio Storti, Fabrizio Balsamo, and Luigi Berio (University of Parma) for their company and the interesting discussions during the Parmelan expedition. Quinten Boersma (Delft University of Technology) is thanked for making available the Brejões orthomosaic and the manual fracture interpretations. The collaboration and scientific inputs of Hilario Bezerra (Universidade Federal do Rio Grande do Norte) during the data acquisition campaign in Brazil is graciously acknowledged. Some maps in this paper were

created using ArcGIS <sup>®</sup> software from ESRI. ArcGIS <sup>®</sup> and ArcMap <sup>™</sup> are the intellectual property of ESRI and are used here under license. Copyright <sup>©</sup> ESRI. All rights reserved. For more information about ESRI <sup>®</sup> software, please visit www.esri.com.

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

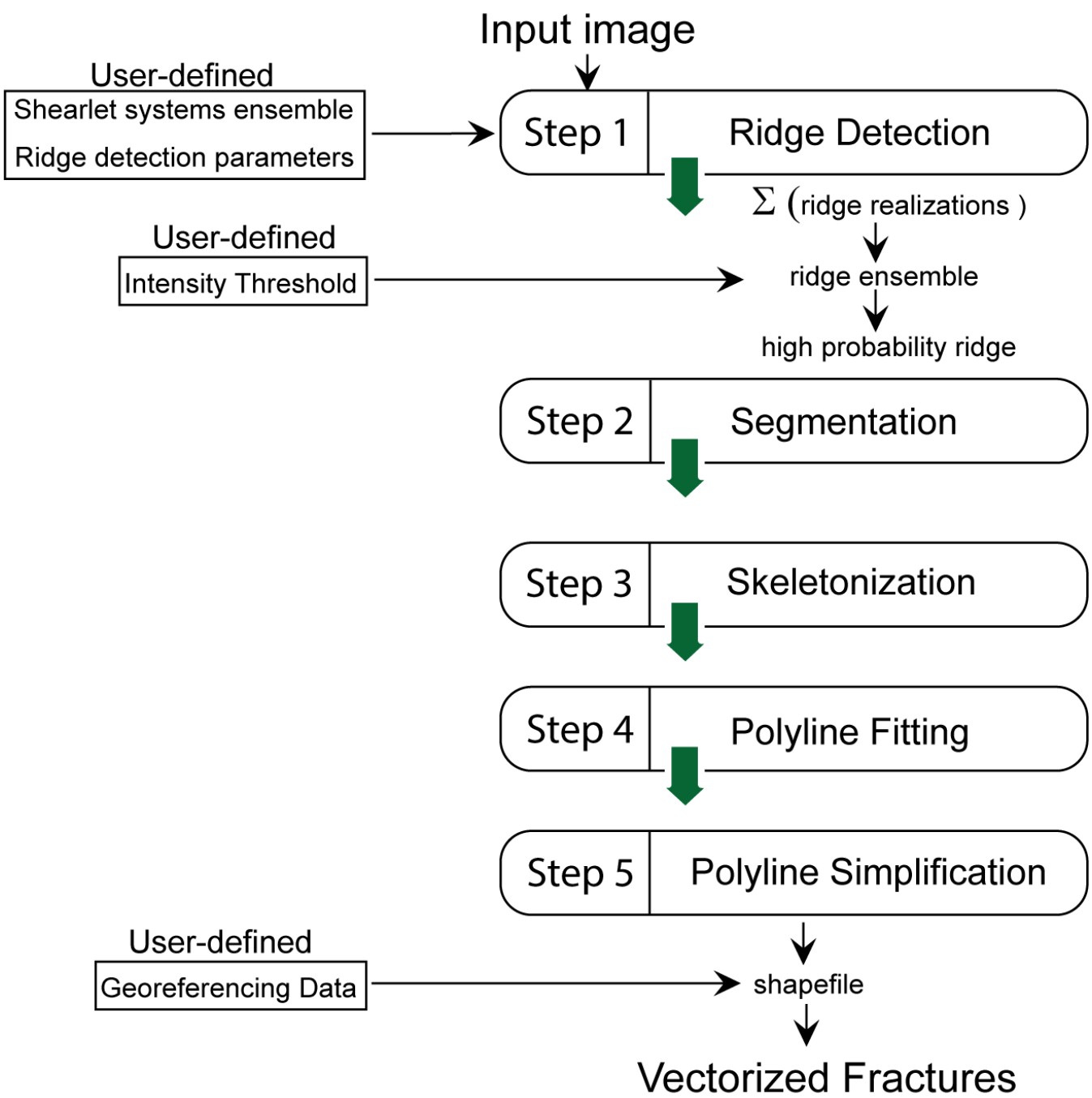

**Figure 1.** The Automated Fracture Trace Detection Workflow

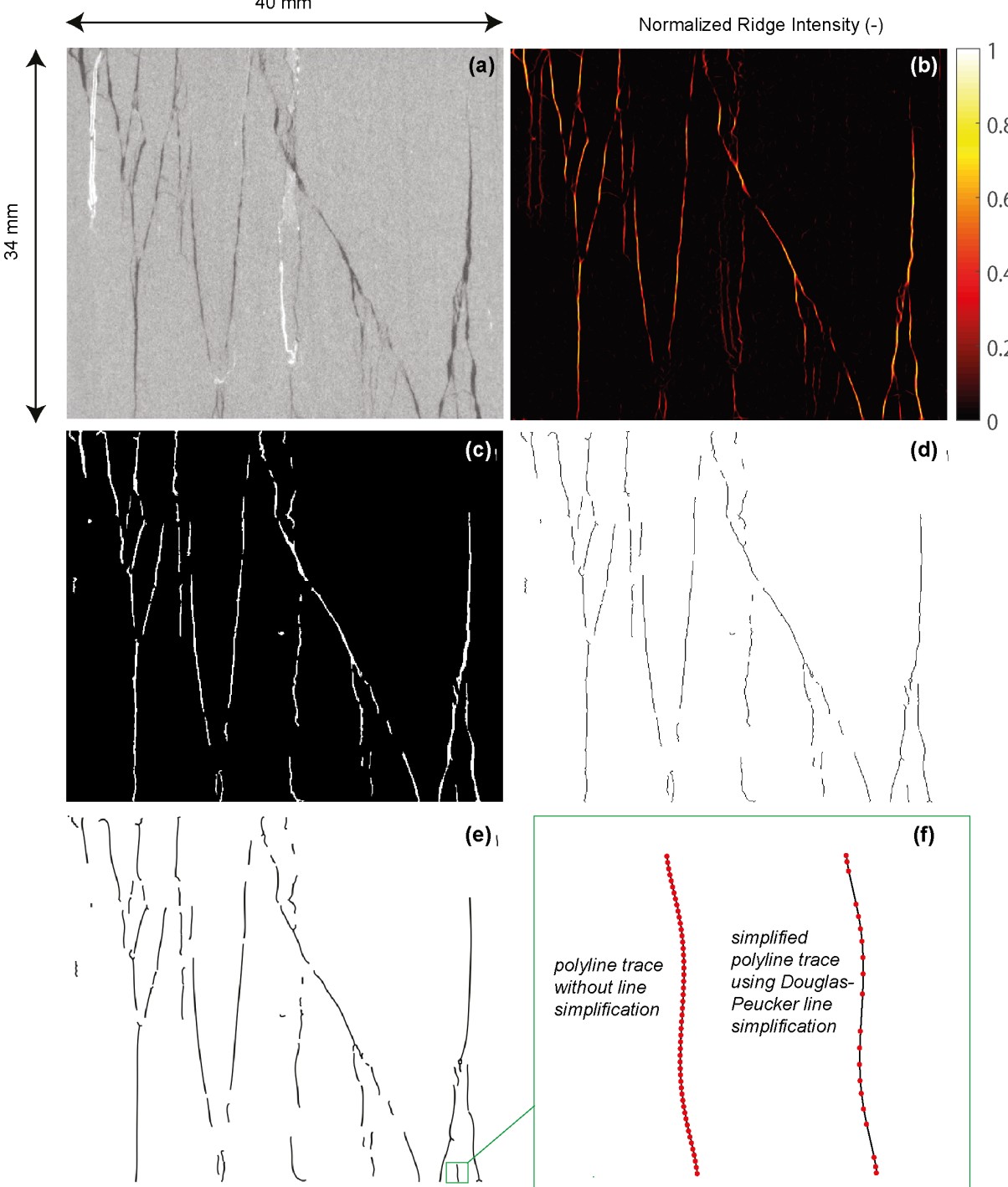

**Figure 2.** Illustration of the steps involved in the automatic fracture extraction using a 40 x 34 mm fractured shale core image (a) CT scan core image from Dwarkasing (2016) (b) Normalized ridge ensemble (c) Segmentation applied on the ridge ensemble (d) Skeletonization applied to the segmented ridge (e) Vectorized polylines fitted to the skeletonized clusters (f) Effect of line simplification applied to a single vectorized segment

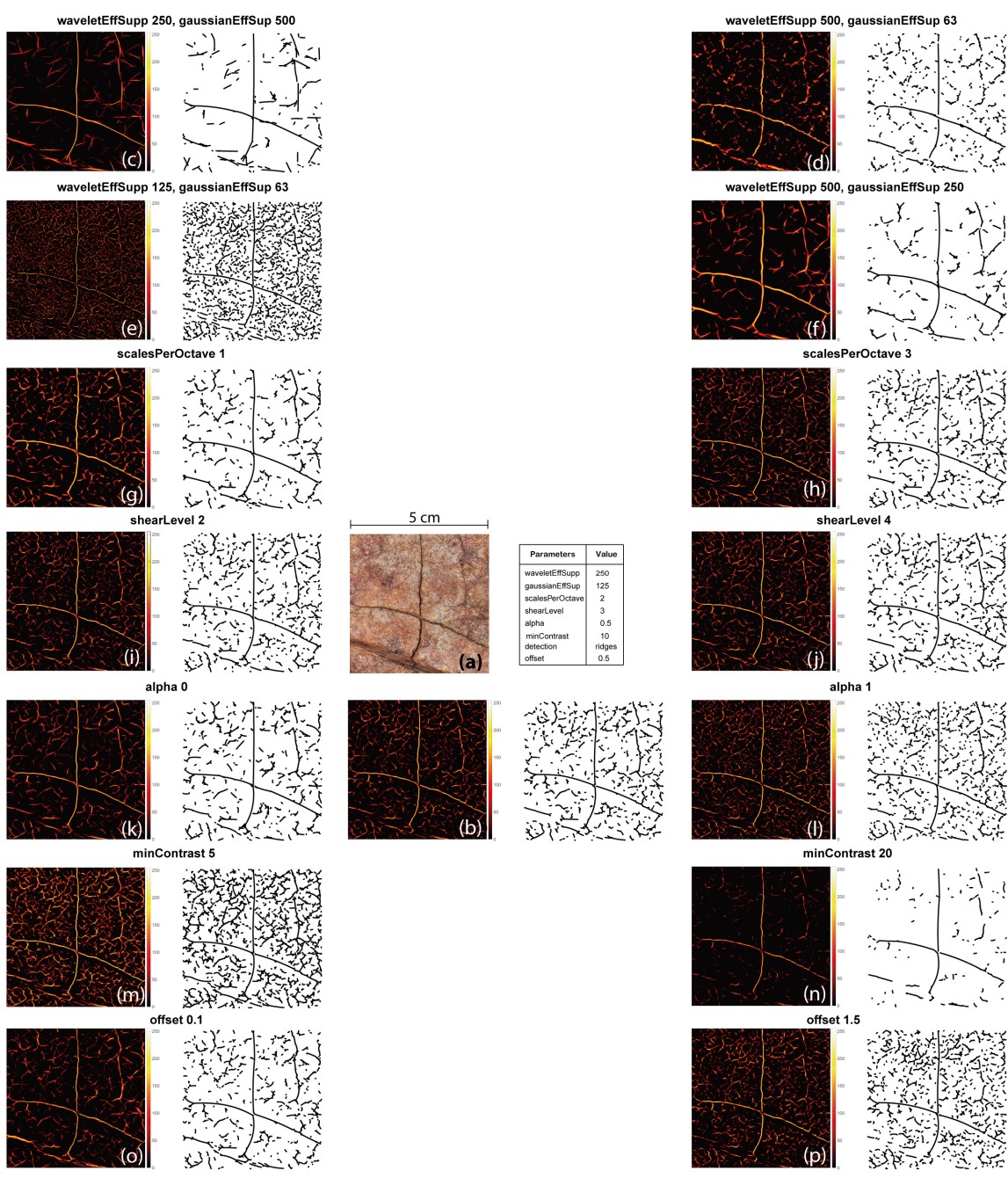

**Figure 3.** Effects of variation of ridge parameters on extracted ridges and the corresponding vectorizations using a fractured siliclastic example. A constant grayscale threshold is applied to the ridge map and all other parameters with respect to post processing are kept constant (a) 5 x 5 cm fractured rock image and base case parameters in table (b) ridge maps and vectorized traces for base case (c) effect of a higher gaussian effect support compared to wavelet support (d) effect of a large difference in wavelet effective support with respect to gaussian support (e),(g),(i),(k),(m),(o) lower bounds of parameters with respect to base case, corresponding ridge maps and traces (f),(h),(j),(l),(n),(p) upper bounds of parameters with respect to base case, corresponding ridge maps and traces

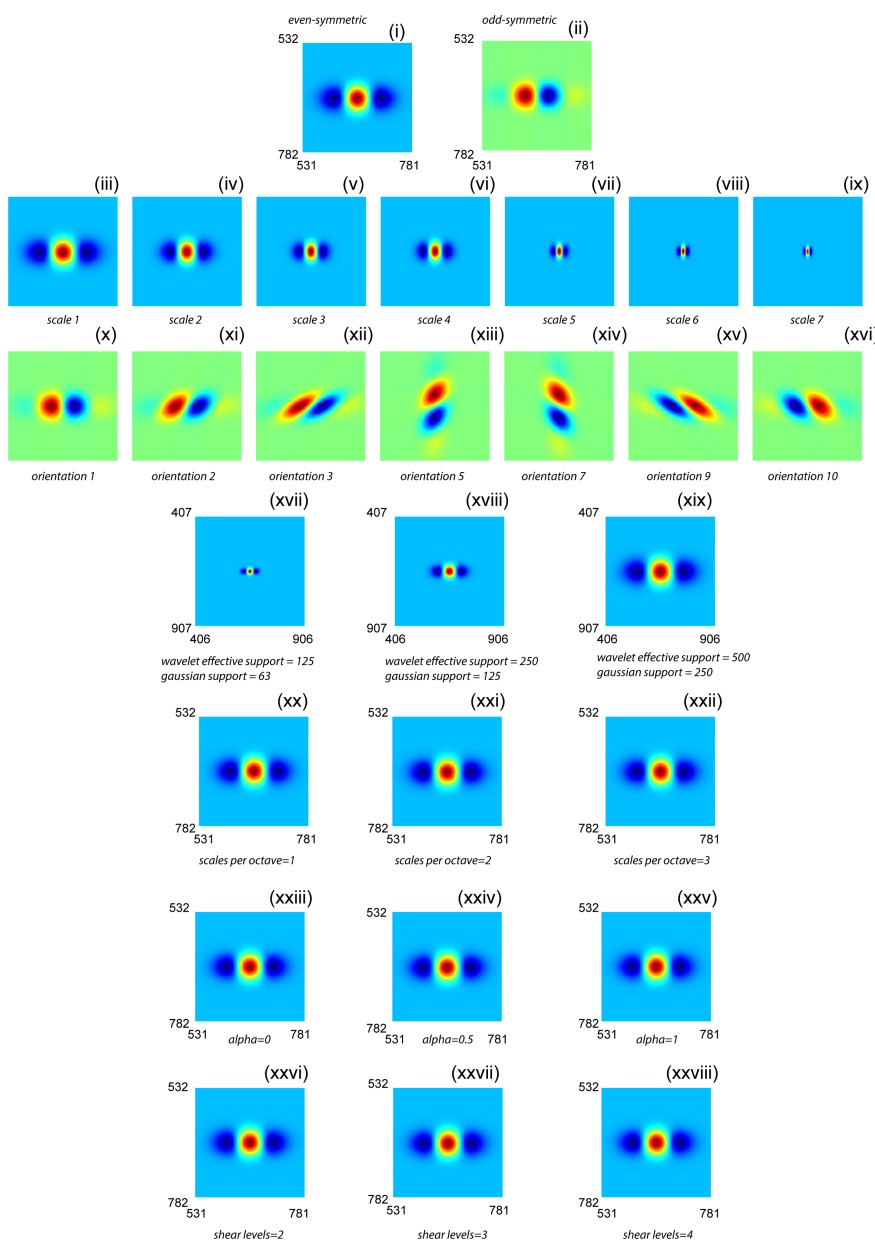

**Figure 4.** Effects of parameter variation on the constructed complex shearlet system for the fractured siliclastic example (i) Even-symmetric elements of the complex shearlet system constructed using the base case parameters in Fig. 3. Full system is 1313 x 1318 pixels. (ii) Odd-symmetric elements of the complex shearlet system using the base case parameters in Fig. 3 (iii) - (ix) depiction of seven scales (x) - (xvi) depiction of seven orientations (out of possible 10) for the odd-symmetric elements of the complex shearlet system (xvii) - (xix) effect of wavelet effective support and gaussian effective support on the even-symmetric elements of the complex shearlet system (xx) effect of gaussian effective support double that of wavelet effective support (xxi) effect of wavelet effective support very large than gaussian effective support (xxii) - (xxiv) effect of scales per octave on the even-symmetric elements (xxv) - (xxvii) effect of anisotropy parameter on the even-symmetric elements (xxviii) - (xxx) effect of shear levels on the even-symmetric elements

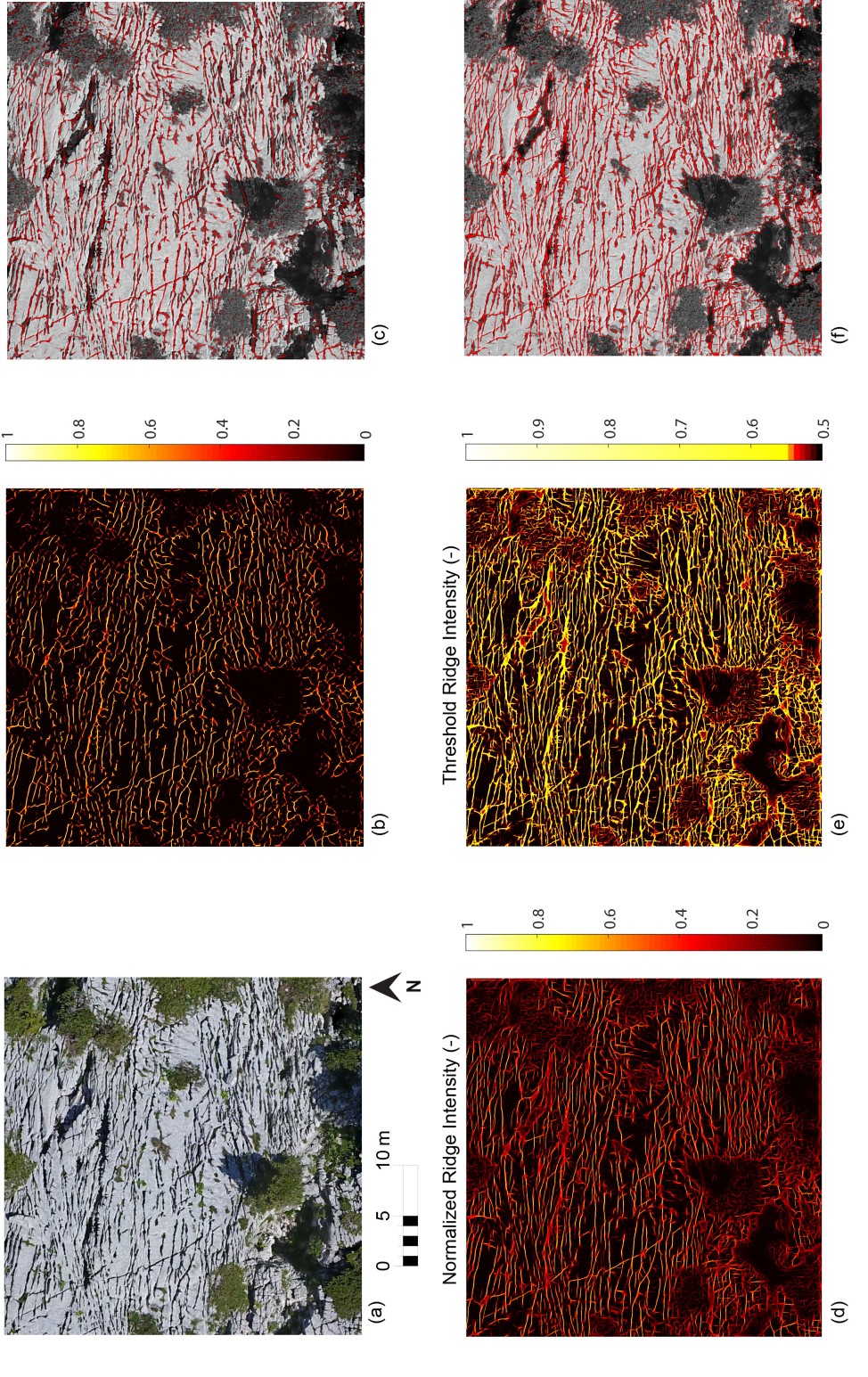

**Figure 5.** Effect of multiple ridge realizations on a sample image from Parmelan Anticline, France (Prabhakaran et al., 2019b) (a) Base Case image used for testing the effect of multiple ridge realizations (b) Ridge map obtained using the base case shearlet parameters in Table. 3 (c) Overlay of ridges obtained using base case shearlet parameters over the test image (d) Normalized ridge intensity ensemble map obtained after 1050 ridge realizations using the parameters in Table. 4 (e) Threshold ridge intensity map that enhances features (f) Overlay of ridges using the threshold ridge intensities over the test image

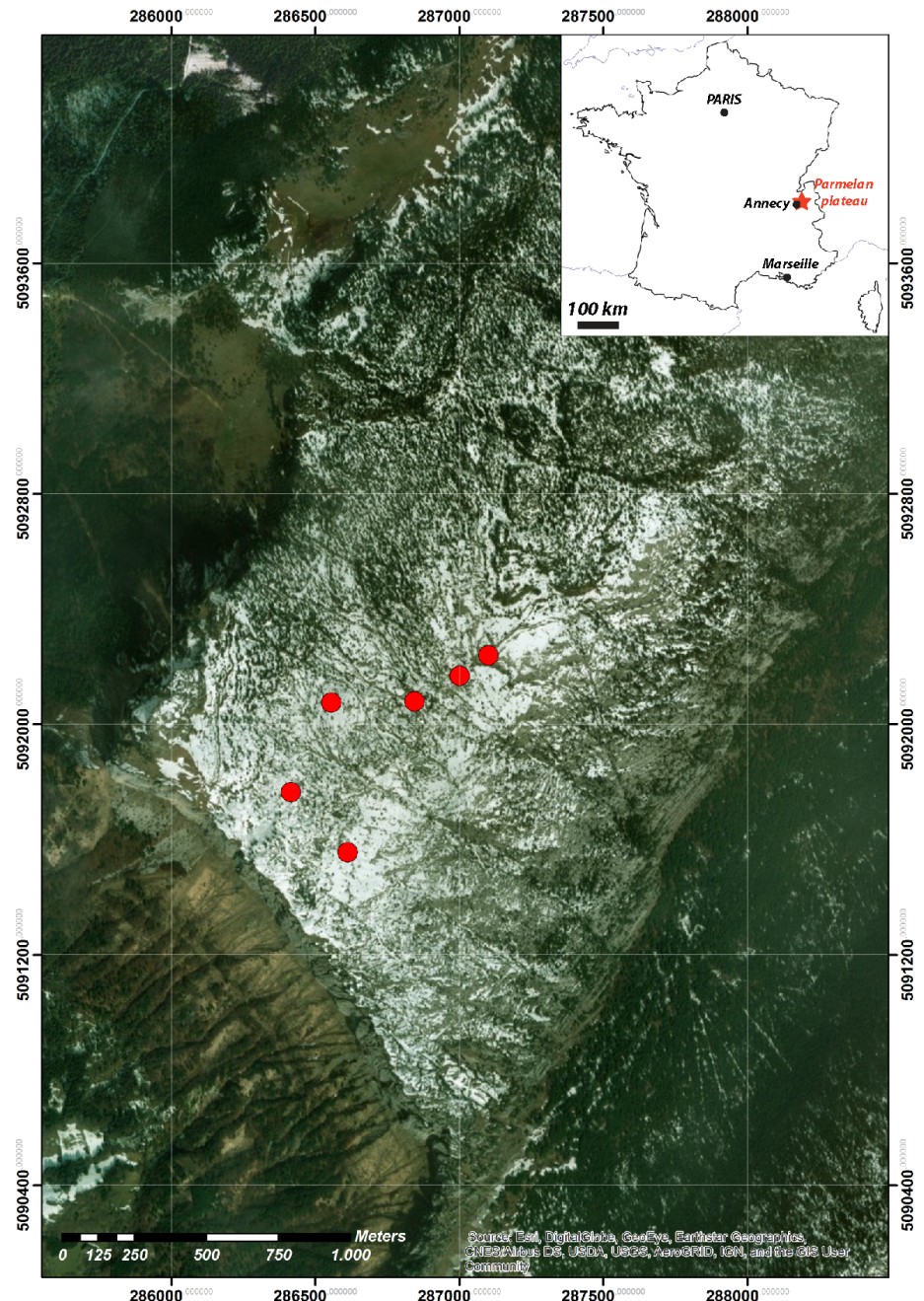

**Figure 6.** Location of the Parmelan plateau in France within the Bornes Massif depicting drone flight base points for six drone missions. This map was generated using satellite imagery obtained from ESRI World Imagery (https://services.arcgisonline.com/ArcGIS/rest/services/ World_Imagery/MapServer) and modified using ArcGIS 10.3, ArcMap 10.3 software by ESRI (http://www.esri.com/). Service Layer Credits: ESRI, DigitalGlobe, GeoEye, Earthstar Geographics, CNES/Airbus DS, USDA, USGS, AeroGRID, IGN, and the GIS User Community.

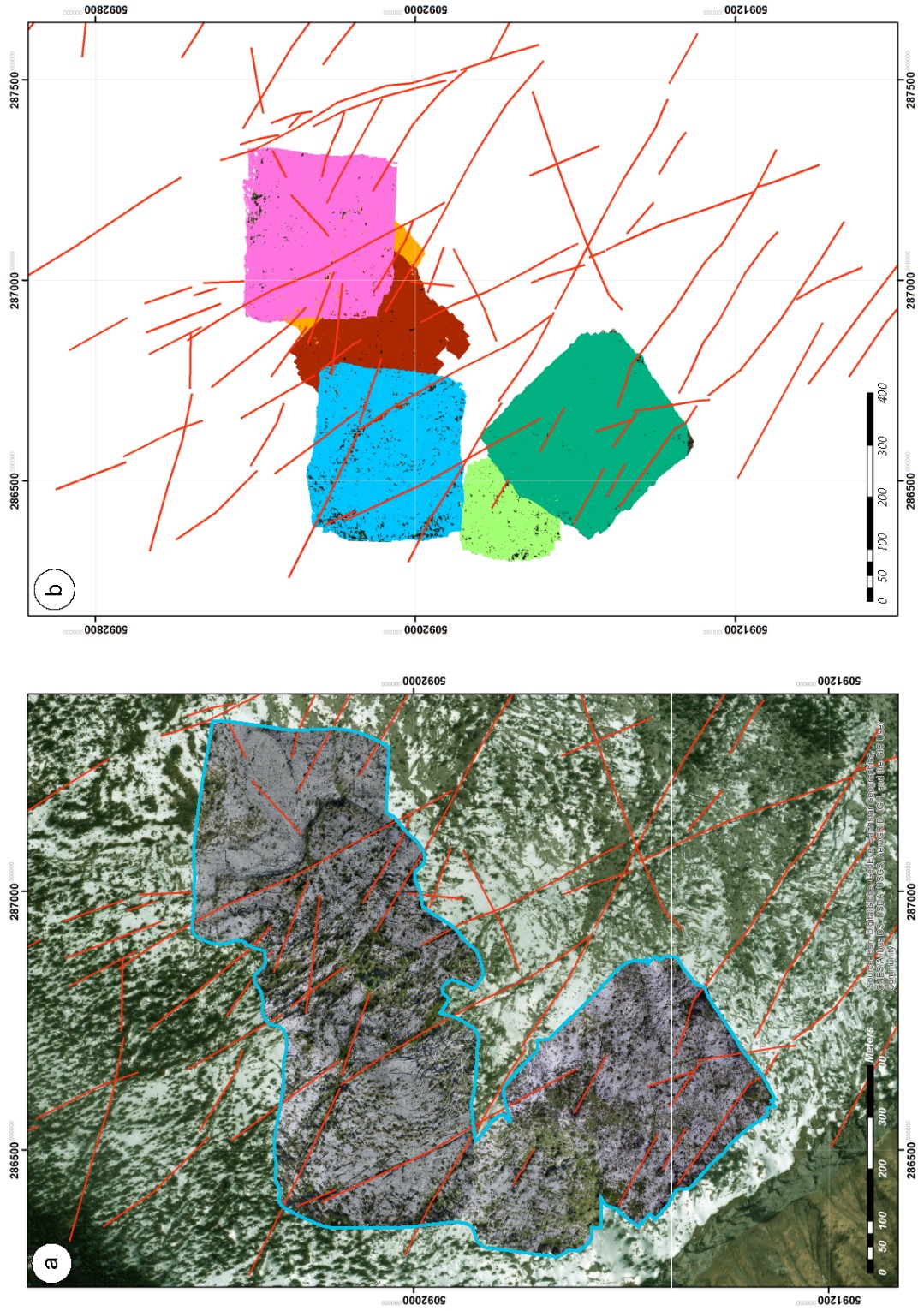

**Figure 7.** Drone photogrammetry coverage area from the Parmelan (a) Region within the Parmelan plateau highlighting the areal extent of the drone photogrammetric orthomosaics which are projected over the base map. Manually traced large scale faults are depicted in red. This map was generated using satellite imagery obtained from ESRI World Imagery (https://services.arcgisonline.com/ArcGIS/rest/services/World_Imagery/MapServer) and modified using ArcGIS 10.3 and ArcMap 10.3 software by ESRI (http://www.esri.com/). Service Layer Credits: ESRI, DigitalGlobe, GeoEye, Earthstar Geographics, CNES/Airbus DS, USDA, USGS, AeroGRID, IGN, and the GIS User Community (b) Spatial extent of the drone coverage of each of the six UAV flight missions in different colours

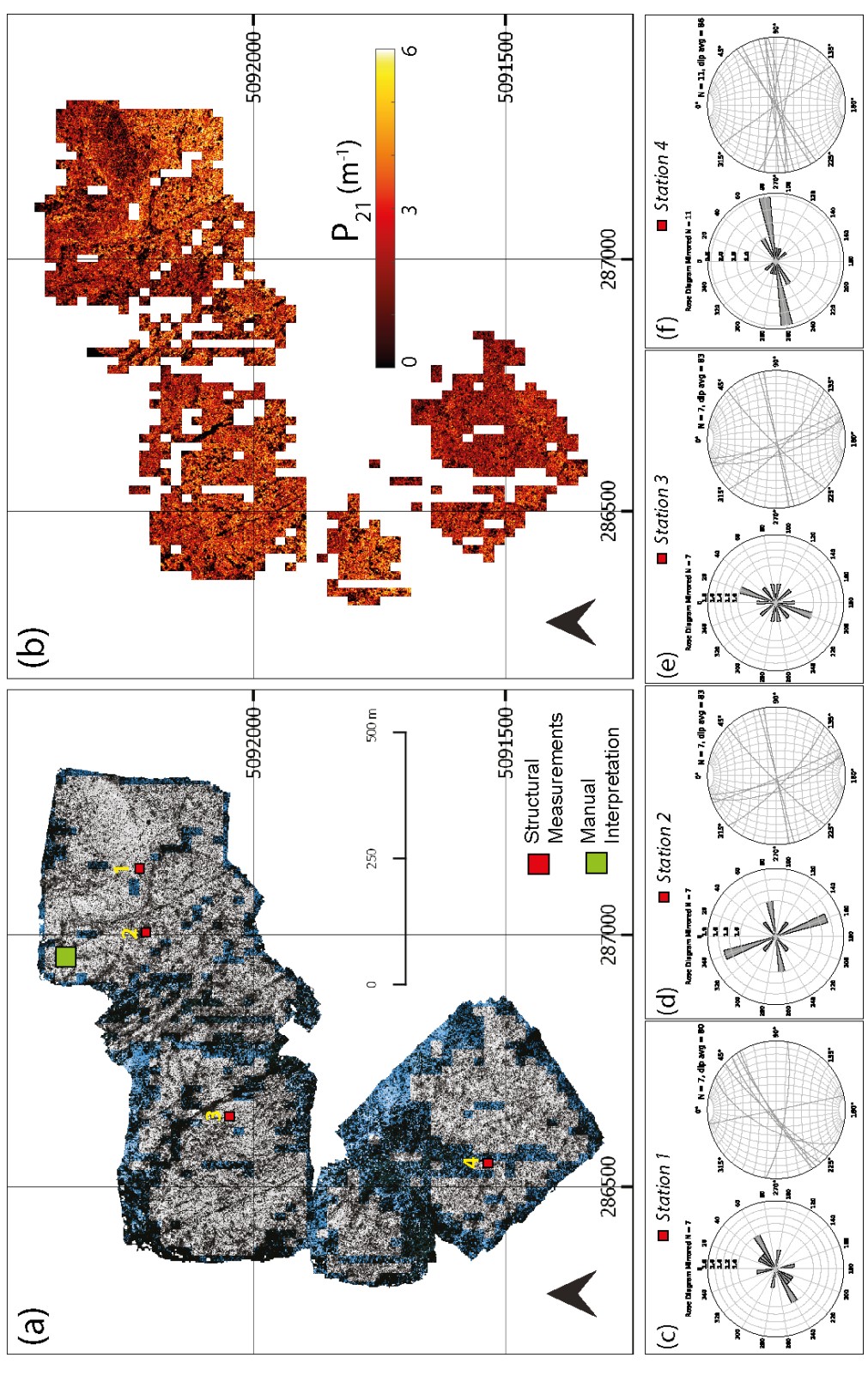

**Figure 8.** Trace extraction results from the Parmelan (a) Selected tiles from the orthomosaic (Prabhakaran et al., 2019b) spatial extent are highlighted. Points where structural measurements were collected (stations) are depicted using red squares. The region where comparison between manual and automatic interpretations is highlighted by the green square (b) Spatial variation of the fracture intensity depicted as a $P_{21}$ plot computed using the box counting method (d) Rose and stereoplot of Station 1 (e) Rose and stereoplot of Station 2 (e) Rose and stereoplot of Station 3 (f) Rose and stereoplot of Station 4. The Parmelan dataset is available at https://doi.org/10.4121/uuid:3f5e255f-edf7-441f-89f2-1adc7ac2f7d1 under a CC-BY-NC-SA license

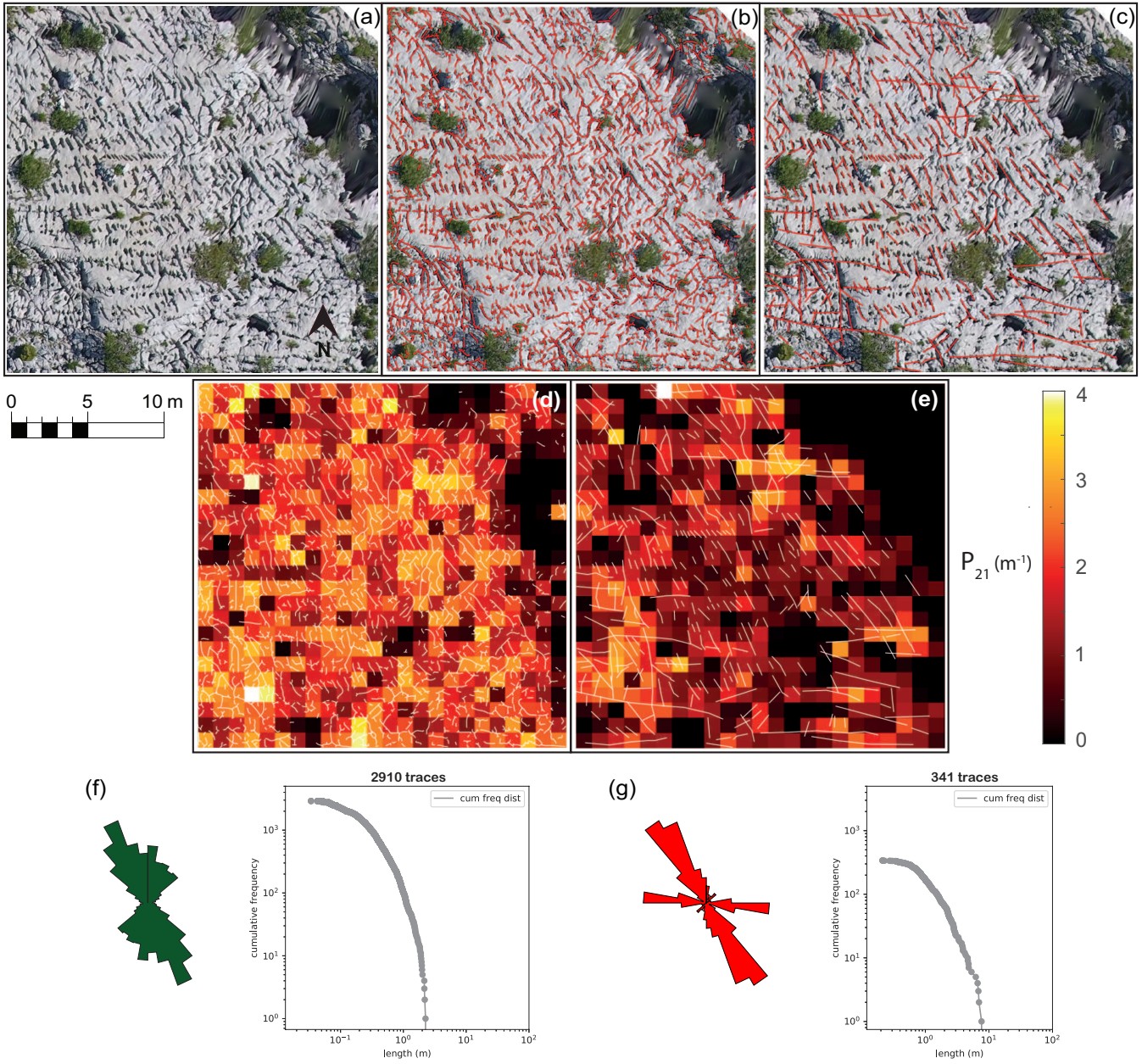

**Figure 9.** Comparison between automatic and manual interpretation on a tile from the Parmelan (a) Tile from the Parmelan orthomosaic (Prabhakaran et al., 2019b) depicting intense fracturing with an organization along the NW-SE corridors (b) Overlay of fractures traced using the automatic detection method (c) Overlay of fractures manually traced for the tile at a zoom of 1:2000 (d) $P_{21}$ Fracture intensity for automatic extracted fractures (e) $P_{21}$ Fracture intensity for manually extracted fractures (f) Length weighted rose plot and cumulative trace length frequency distribution for the automated result (g) Length weighted rose plot and cumulative trace length frequency distribution for the manual result. The Parmelan dataset is available at https://doi.org/10.4121/uuid:3f5e255f-edf7-441f-89f2-1adc7ac2f7d1 under a CC-BY-NC-SA license

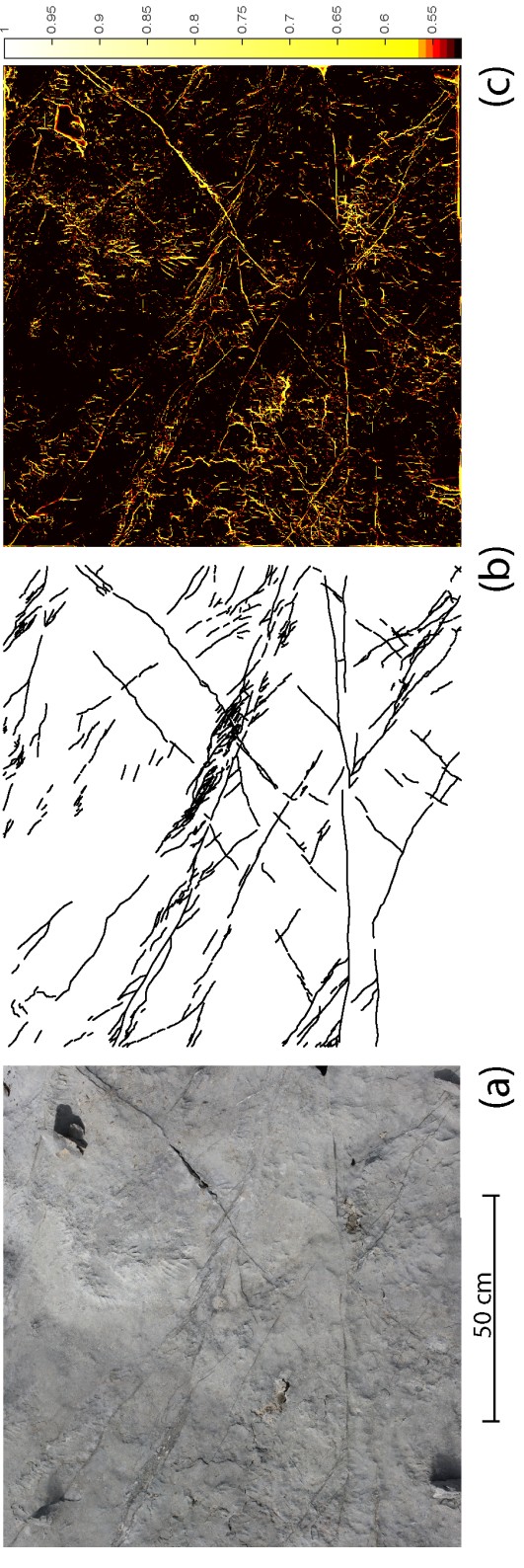

**Figure 10.** Extension of the automated method to extract mineralized fractures (a) Image from Parmelan depicting mineralized fractures (b) Manual interpretation of mineralized fractures (c) Ridge ensemble

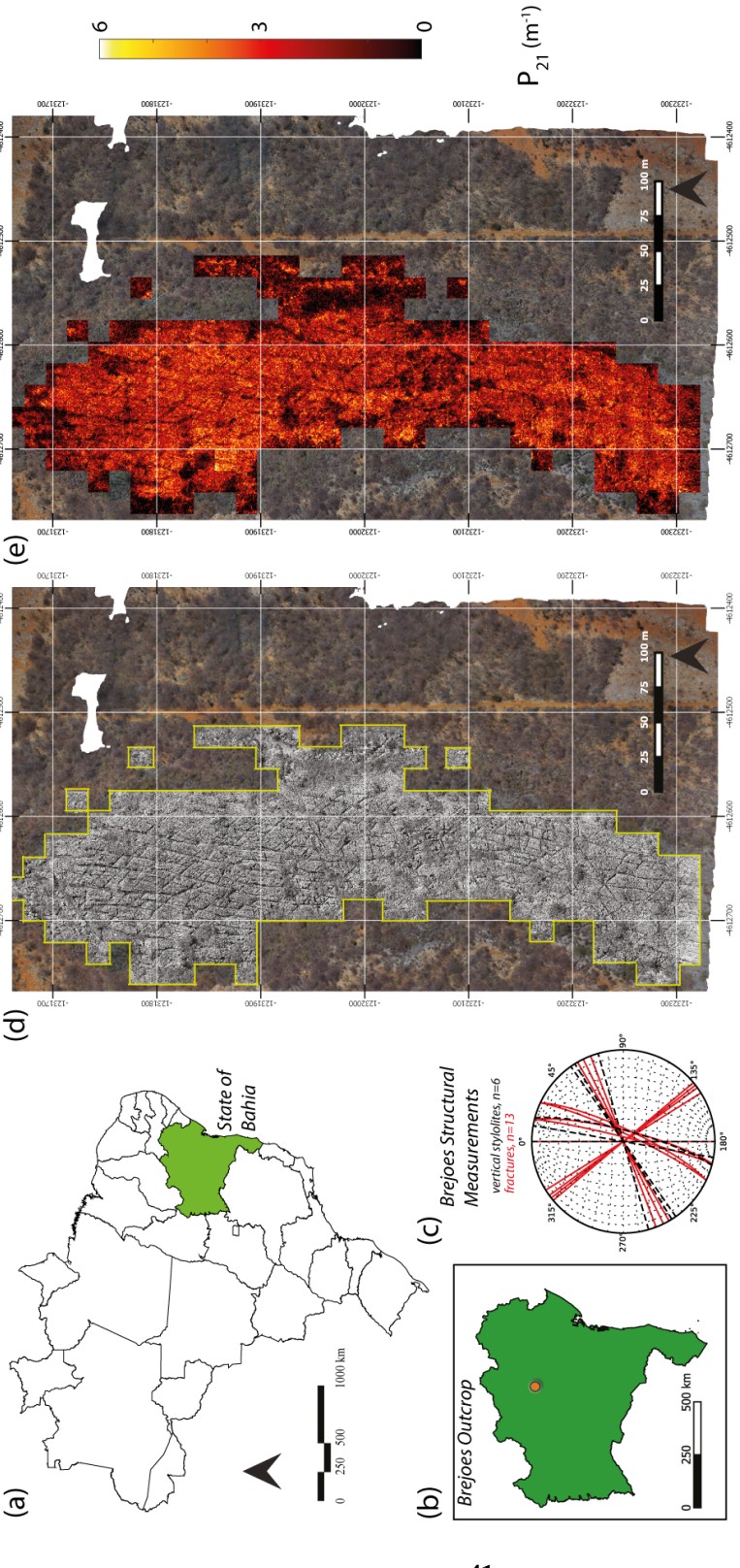

**Figure 11.** Trace extraction results from the Brejões outcrop (a) Bahia state in NE Brazil (b) Location of the Brejões outcrop in the state of Bahia. (c) Structural measurements from Brejões outcrop, adapted from Boersma et al. (2019) (d) Selected tiles from the Brejões orthomosaic (Prabhakaran et al., 2019a) for the automated extraction (e) Spatial variation of the fracture intensity depicted as a $P_{21}$ plot computed using the box counting method. Maps depicting administrative boundaries of Brazil and Bahia state was modified from free vector spatial data downloadable at DIVA-GIS (https://www.diva-gis.org/Data). The Brejões dataset is available at https://doi.org/10.4121/uuid:67cde05c-9e99-4cc4-8cec-9f2666457d1f under a CC-BY-NC-SA license

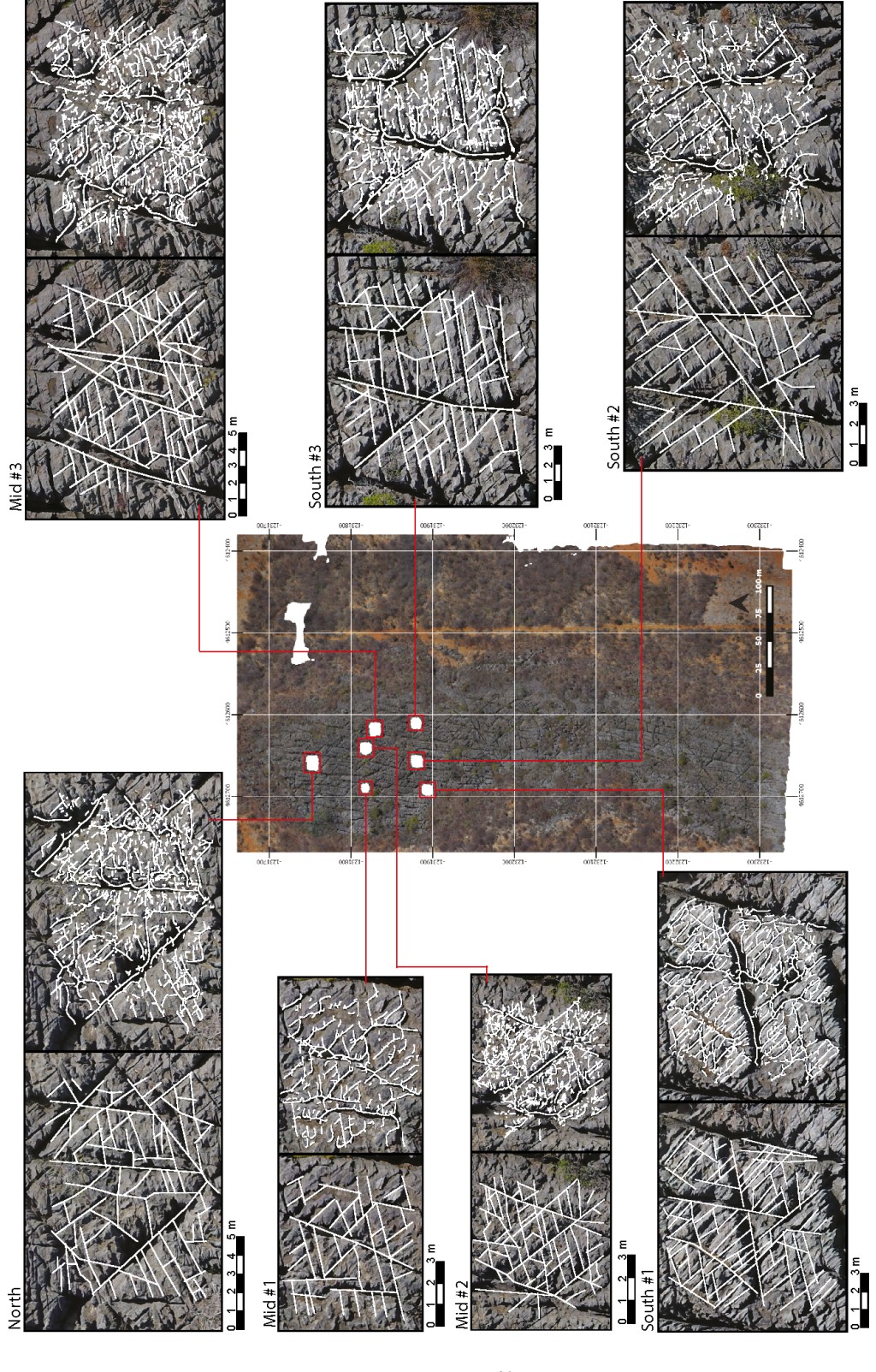

**Figure 12.** Comparison between manual (left) and automatic (right) interpretation on seven stations within the Brejões outcrop (Prabhakaran et al., 2019a). The manual interpretations were obtained from Boersma et al. (2019). The Brejões dataset is available at https://doi.org/10.4121/uuid: 67cde05c-9e99-4cc4-8cec-9f2666457d1f under a CC-BY-NC-SA license

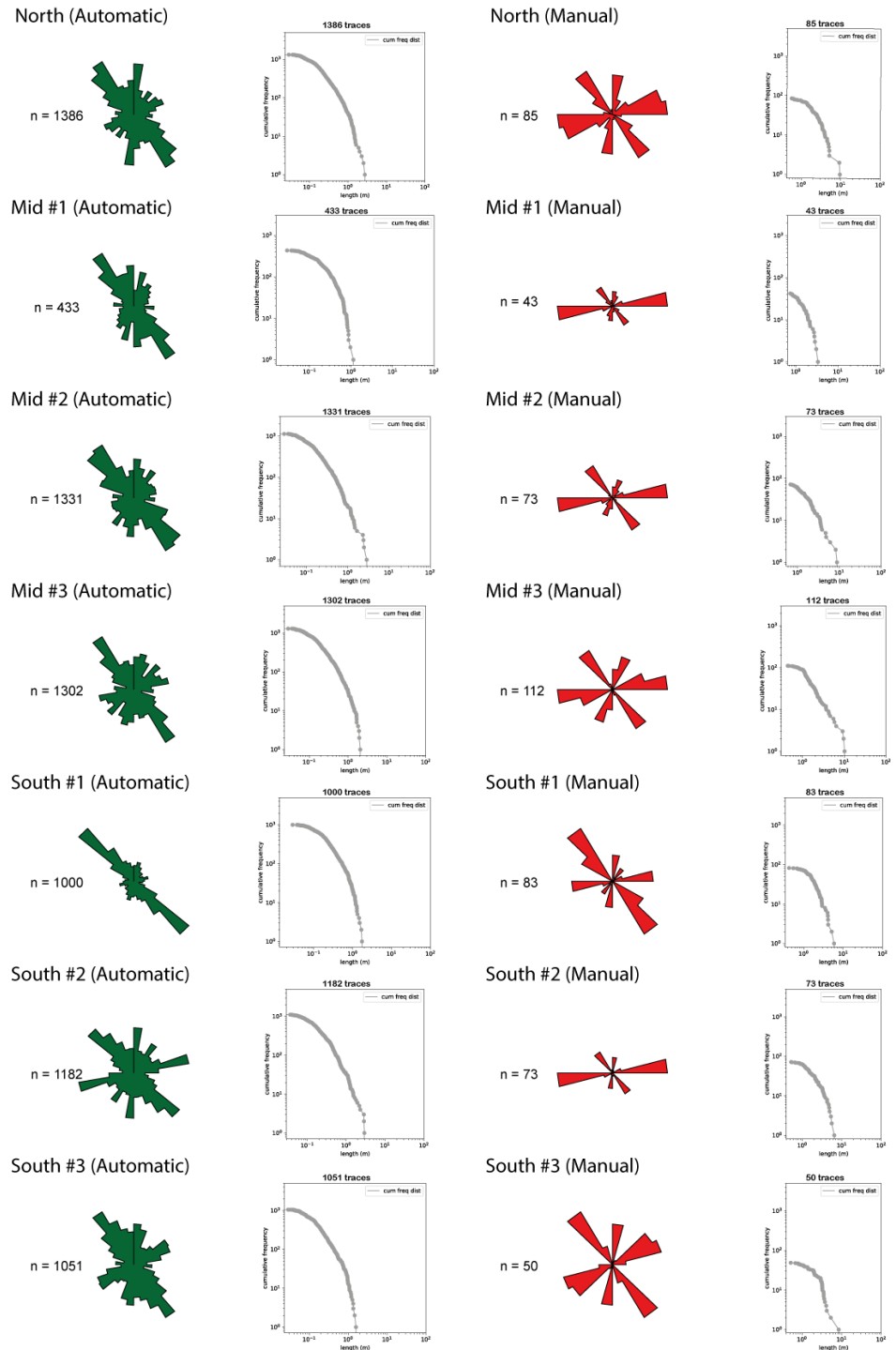

**Figure 13.** Comparison of trace length weighted rose plots and cumulative trace length distributions for automatic and manual trace interpretations from Brejões outcrop stations. The Brejões dataset is available at https://doi.org/10.4121/uuid: 67cde05c-9e99-4cc4-8cec-9f2666457d1f under a CC-BY-NC-SA license

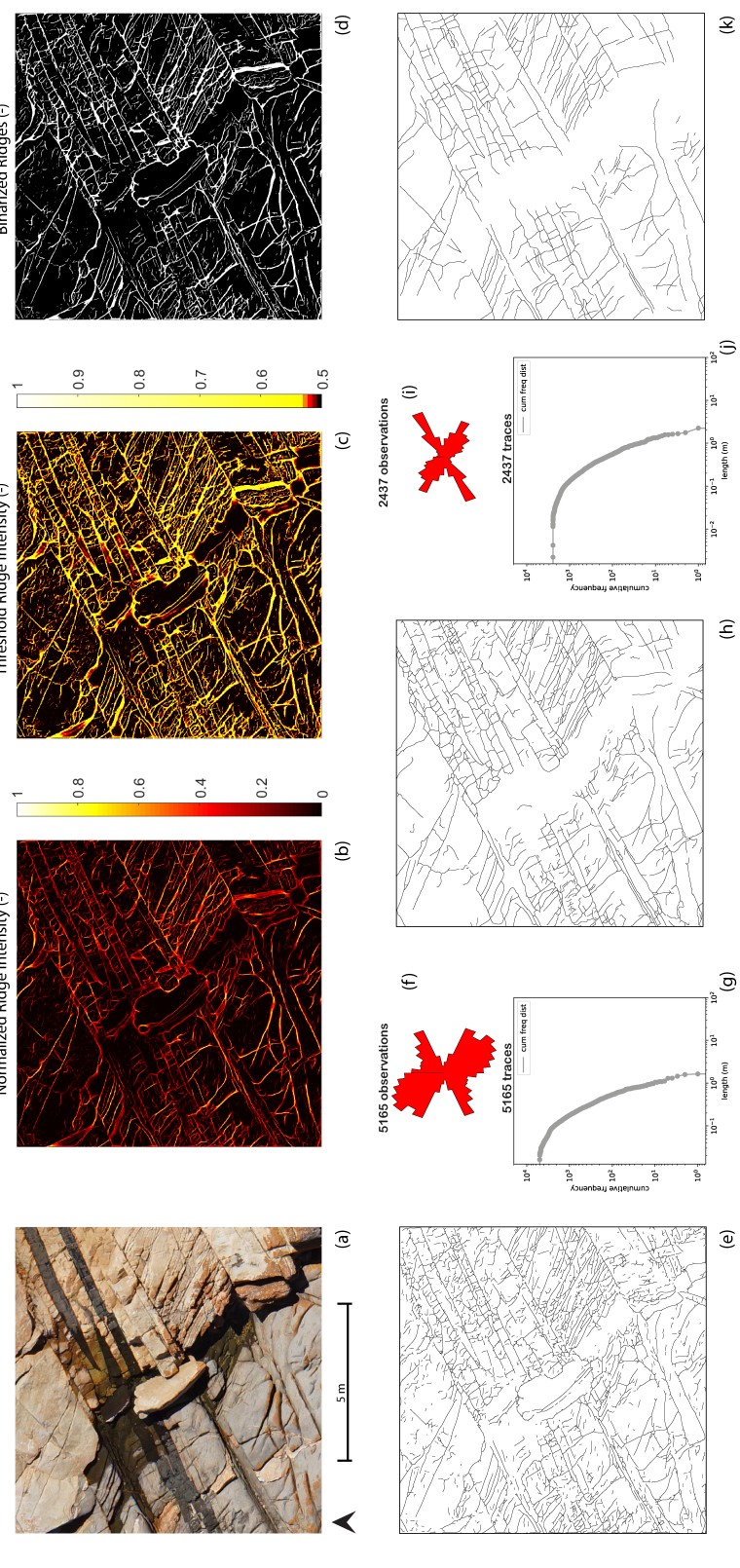

**Figure 14.** Comparison of Benchmark Image 1 (a) Bingie Bingie Area 1 from Thiele et al. (2017). This image is available for download at https://doi.org/10. 4225/03/5981b31091af9 under a CC-BY-4.0 license (b) Normalized ridge map using complex shearlet automatic extraction (c) Threshold applied to the normalized ridges (d) Binarized ridges map (e) Vectorized automatic traces (f) Length weighted rose plot of the automatic interpreted traces (g) Cumulative frequency plot of the automatic interpreted traces (h) Assisted cleaned up trace map for Area 1 (i) Length weighted rose plot of the assisted traces (j) Cumulative frequency plot of the assisted traces (k) Assisted trace interpretation modified from Thiele et al. (2017) for comparison.

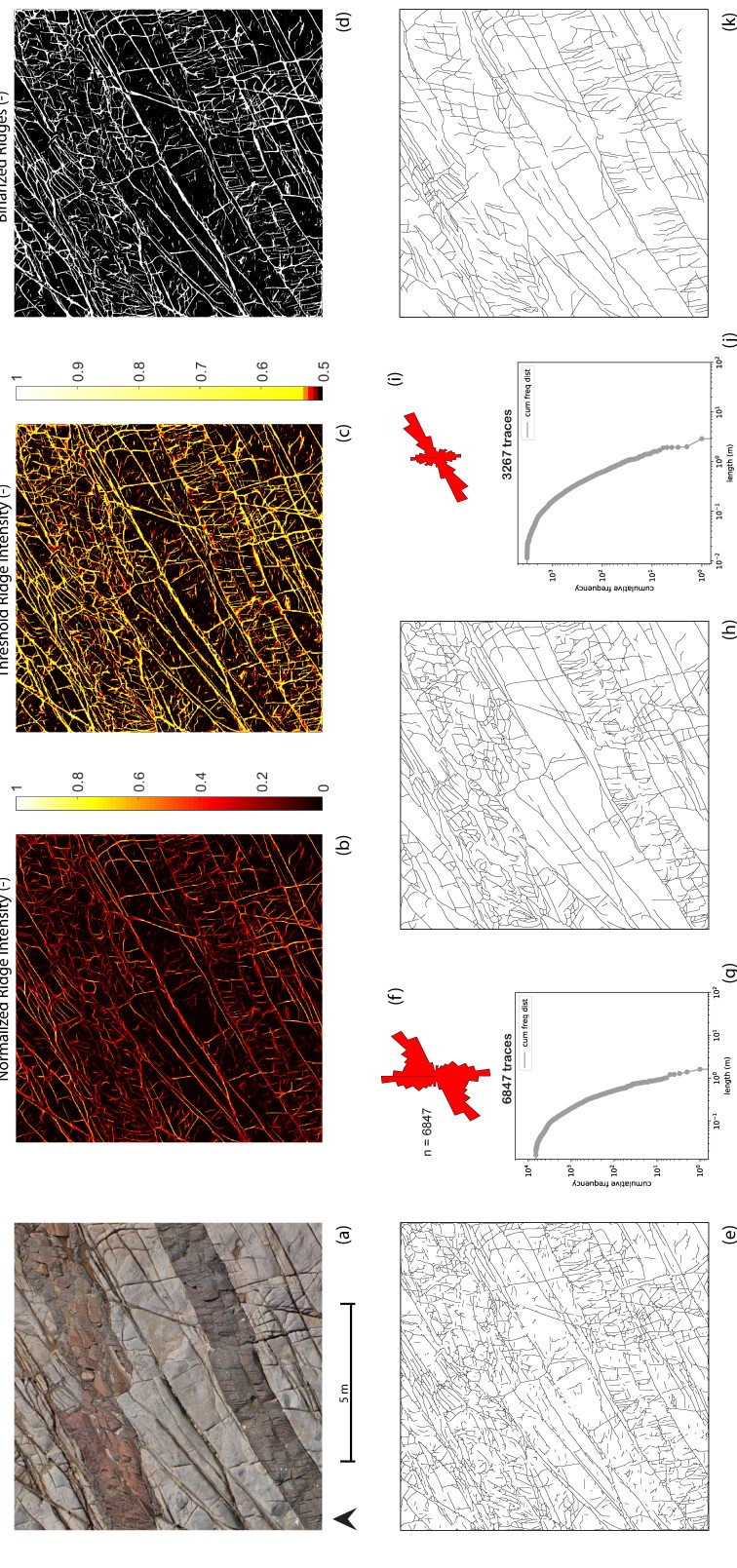

**Figure 15.** Comparison of Benchmark Image 2 (a) Bingie Bingie Area 2 from Thiele et al. (2017). This image is available for download at https://doi.org/10. 4225/03/5981b31091af9 under a CC-BY-4.0 license (b) Normalized ridge map using complex shearlet automatic extraction (c) Threshold applied to the normalized ridges (d) Binarized ridges map (e) Vectorized automatic traces (f) Length weighted rose plot of the automatic interpreted traces (g) Cumulative frequency plot of the automatic interpreted traces (h) Assisted cleaned up trace map for Area 2 (i) Length weighted rose plot of the assisted traces (j) Cumulative frequency plot of the assisted traces (k) Assisted trace interpretation modified from Thiele et al. (2017) for comparison

**Table 1.** Shearlet System Parameters

| Parameter | Description |
| --- | --- |
| *waveletEffSupp* | Length of the effective support in pixels of the wavelet |
| *gaussianEffSupp* | Length of the effective support in pixels of the Gaussian filter |
| *scalesPerOctave* | Number of intermediate scales for each octave |
| *shearLevel* | Number of differently oriented shearlets on each scale |
| *alpha* | Degree of anisotropy introduced via scaling |
| *octaves* | Number of octaves spanned by the shearlet system |

**Table 2.** Detection Parameters

| Parameter | Description |
| --- | --- |
| *DetectionType* | Specification of detection measure (edge/ridge) |
| *minContrast* | Specification of the level of contrast for edge/ridge detection |
| *offset* | Scaling offset between the even- and odd- symmetric shearlets |

**Table 3.** Shearlet system and detection parameters used to extract ridges for the base case

| Base Case Parameters | |
| --- | --- |
| *waveletEffSupp* | 125 |
| *gaussianEffSup* | 63 |
| *scalesPerOctave* | 2 |
| *shearLevel* | 3 |
| *alpha* | 0.5 |
| *octaves* | 3.5 |
| *minContrast* | 10 |
| *detection* | negative ridges |
| *even/odd offset* | 1 |

**Table 4.** Ensemble for Parameter Variation

| Parameter | Values |
| --- | --- |
| *waveletEffSupp* | Original image size in pixels divided by 5, 8, 10, 12 and 15 |
| *gaussianEffSupp* | Original image size in pixels divided by 5, 8, 10, 12 and 15 |
| *scalesPerOctave* | 1,2,3 and 4 |
| *shearLevel* | 2,3 and 4 |
| *alpha* | 0, 0.25, 0.5, 0.75, 1 |
| *minContrast* | 1, 5, 10, 15, 20 |
| *even/odd offset* | 0.001, 0.01, 0.1, 1, 2 |

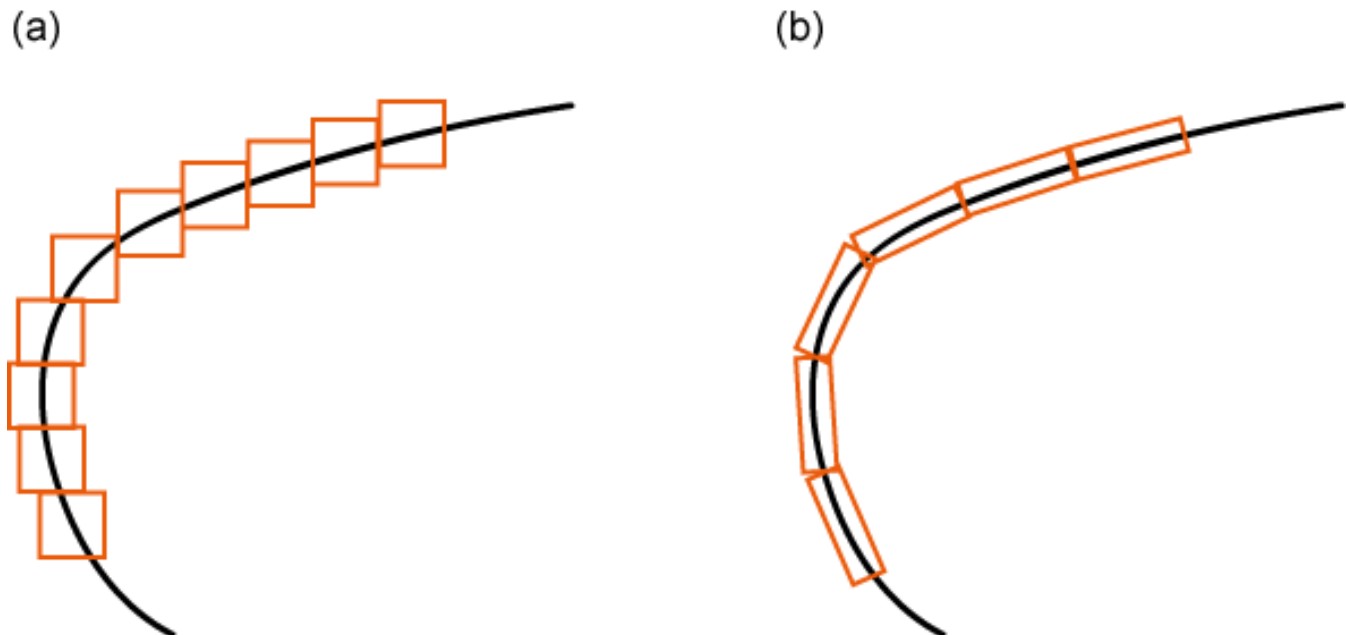

**Figure A1.** (a). Isotropic elements capturing a discontinuity curve (b). Sheared, anisotropic elements capturing a discontinuity curve (modified from Kutyniok and Labate, 2012)

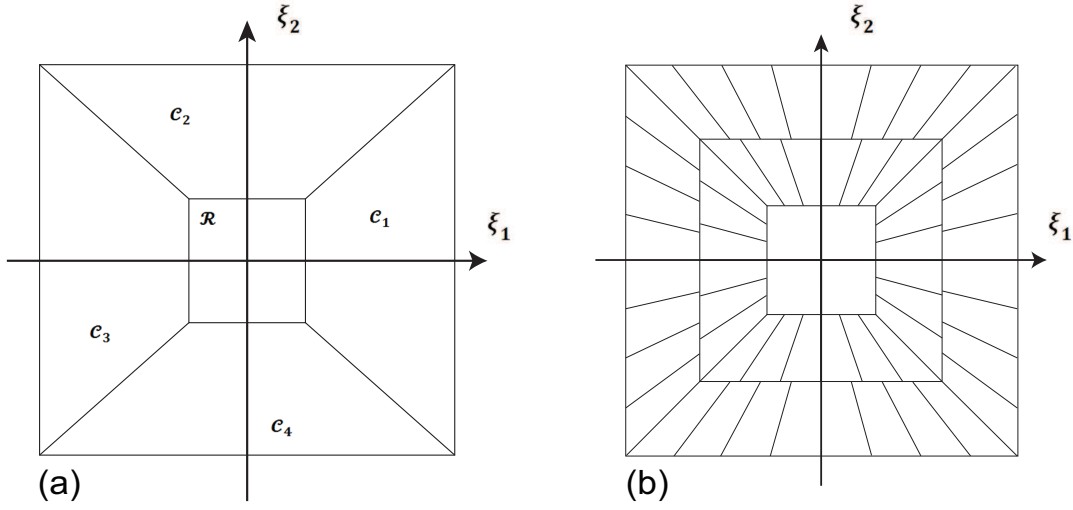

**Figure A2.** The cone adapted continuous shearlet system (a) Bias in directions is handled by dividing the frequency plane into 4 cones $\mathcal{C}_1$, $\mathcal{C}_2$, $\mathcal{C}_3$, $\mathcal{C}_4$ and a square low frequency box region in the centre $\mathcal{R}$ (b) Trapezoidal shaped wedge tiling of the frequency induced domain induced by the shearlet transform (modified after Kutyniok and Labate 2012)

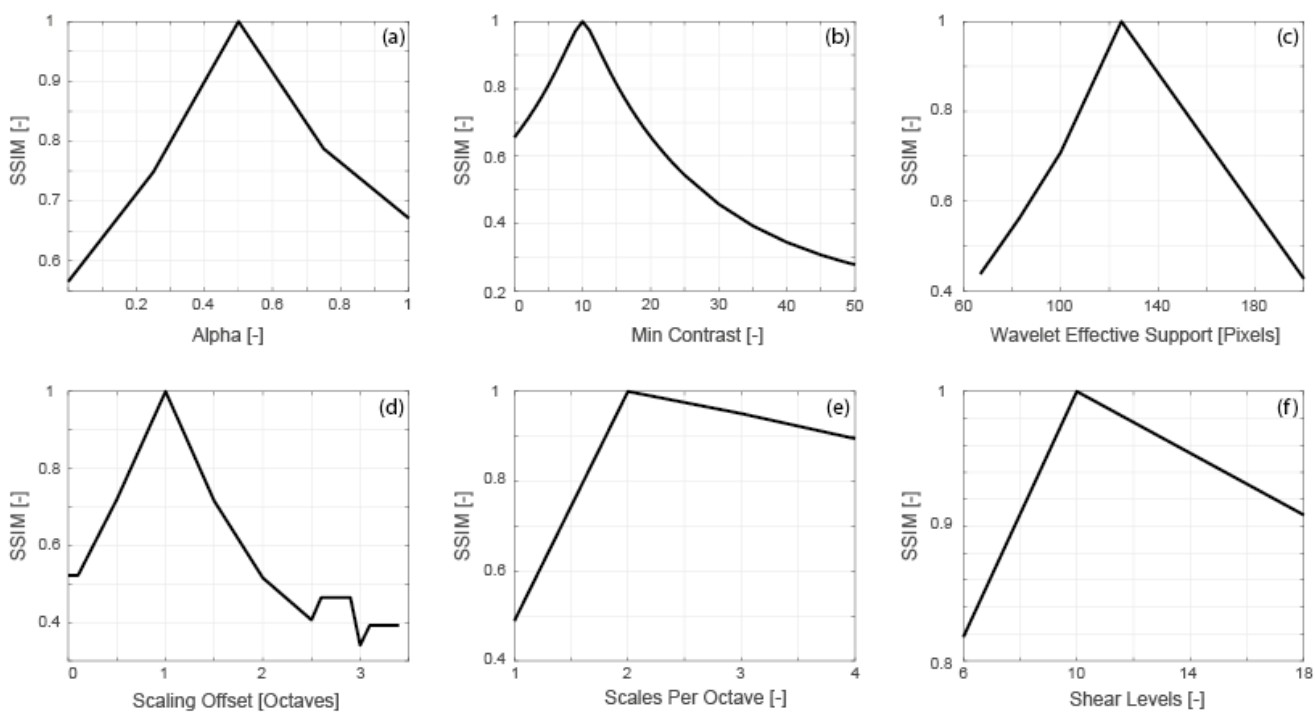

**Figure A3.** Variation of the Structural Similarity Index of of base case ridges with shearlet and detection parameters (a) SSIM vs Anisotropy Exponent (b) SSIM vs MinContrast (c) SSIM vs Wavelet Effective Support (d) SSIM vs Scaling Offset (e) SSIM vs Scales / Octave (f) SSIM vs Shearlevels

**Table A1.** Shearlets

| Shearlet System | waveletEffSupp | gaussianEffSupp | scalesPerOctave | shearLevel | scales | nShearlets | alpha |
|---|---|---|---|---|---|---|---|
| 1 | 200 | 100 | 1 | 2 | 1 | 36 | 0 |
| 2 | 125 | 63 | 1 | 2 | 1 | 36 | 0 |
| 3 | 84 | 42 | 1 | 2 | 1 | 36 | 0 |
| 4 | 67 | 34 | 1 | 2 | 1 | 36 | 0 |
| 5 | 200 | 100 | 2 | 2 | 7 | 12 | 0 |
| 6 | 125 | 63 | 2 | 2 | 7 | 12 | 0 |
| 7 | 84 | 42 | 2 | 2 | 7 | 12 | 0 |
| 8 | 67 | 34 | 2 | 2 | 7 | 12 | 0 |
| 9 | 200 | 100 | 3 | 2 | 10.5 | 12 | 0 |
| 10 | 125 | 63 | 3 | 2 | 10.5 | 12 | 0 |
| 11 | 84 | 42 | 3 | 2 | 10.5 | 12 | 0 |
| 12 | 67 | 34 | 3 | 2 | 10.5 | 12 | 0 |
| 13 | 200 | 100 | 4 | 2 | 14 | 12 | 0 |
| 14 | 125 | 63 | 4 | 2 | 14 | 12 | 0 |
| 15 | 84 | 42 | 4 | 2 | 14 | 12 | 0 |
| 16 | 67 | 34 | 4 | 2 | 14 | 12 | 0 |
| 17 | 200 | 100 | 1 | 3 | 3.5 | 20 | 0 |
| 18 | 125 | 63 | 1 | 3 | 3.5 | 20 | 0 |
| 19 | 84 | 42 | 1 | 3 | 3.5 | 20 | 0 |
| 20 | 67 | 34 | 1 | 3 | 3.5 | 20 | 0 |
| 21 | 200 | 100 | 2 | 3 | 7 | 20 | 0 |
| 22 | 125 | 63 | 2 | 3 | 7 | 20 | 0 |
| 23 | 84 | 42 | 2 | 3 | 7 | 20 | 0 |
| 24 | 67 | 34 | 2 | 3 | 7 | 20 | 0 |
| 25 | 200 | 100 | 3 | 3 | 10.5 | 20 | 0 |
| 26 | 125 | 63 | 3 | 3 | 10.5 | 20 | 0 |
| 27 | 84 | 42 | 3 | 3 | 10.5 | 20 | 0 |
| 28 | 67 | 34 | 3 | 3 | 10.5 | 20 | 0 |
| 29 | 200 | 100 | 4 | 3 | 14 | 20 | 0 |
| 30 | 125 | 63 | 4 | 3 | 14 | 20 | 0 |
| 31 | 84 | 42 | 4 | 3 | 14 | 20 | 0 |
| 32 | 67 | 34 | 4 | 3 | 14 | 20 | 0 |
| 33 | 200 | 100 | 1 | 4 | 3.5 | 36 | 0 |
| 34 | 125 | 63 | 1 | 4 | 3.5 | 36 | 0 |
| 35 | 84 | 42 | 1 | 4 | 3.5 | 36 | 0 |
| 36 | 67 | 34 | 1 | 4 | 3.5 | 36 | 0 |
| 37 | 200 | 100 | 2 | 4 | 7 | 36 | 0 |
| 38 | 125 | 63 | 2 | 4 | 7 | 36 | 0 |
| 39 | 84 | 42 | 2 | 4 | 7 | 36 | 0 |
| 40 | 67 | 34 | 2 | 4 | 7 | 36 | 0 |
| 41 | 200 | 100 | 3 | 4 | 10.5 | 36 | 0 |
| 42 | 125 | 63 | 3 | 4 | 10.5 | 36 | 0 |
| 43 | 84 | 42 | 3 | 4 | 10.5 | 36 | 0 |
| 44 | 67 | 34 | 3 | 4 | 10.5 | 36 | 0 |
| 45 | 200 | 100 | 4 | 4 | 14 | 36 | 0 |
| 46 | 125 | 63 | 4 | 4 | 14 | 36 | 0 |
| 47 | 84 | 42 | 4 | 4 | 14 | 36 | 0 |
| 48 | 67 | 34 | 4 | 4 | 14 | 36 | 0 |
| 49 | 200 | 100 | 1 | 2 | 3.5 | 12 | 0.5 |
| 50 | 125 | 63 | 1 | 2 | 3.5 | 12 | 0.5 |
| 51 | 84 | 42 | 1 | 2 | 3.5 | 12 | 0.5 |
| 52 | 67 | 34 | 1 | 2 | 3.5 | 12 | 0.5 |
| 53 | 200 | 100 | 2 | 2 | 7 | 12 | 0.5 |
| 54 | 125 | 63 | 2 | 2 | 7 | 12 | 0.5 |
| 55 | 84 | 42 | 2 | 2 | 7 | 12 | 0.5 |
| 56 | 67 | 34 | 2 | 2 | 7 | 12 | 0.5 |
| 57 | 200 | 100 | 3 | 2 | 10.5 | 12 | 0.5 |
| 58 | 125 | 63 | 3 | 2 | 10.5 | 12 | 0.5 |
| 59 | 84 | 42 | 3 | 2 | 10.5 | 12 | 0.5 |
| 60 | 67 | 34 | 3 | 2 | 10.5 | 12 | 0.5 |
| 61 | 200 | 100 | 4 | 2 | 14 | 12 | 0.5 |
| 62 | 125 | 63 | 4 | 2 | 14 | 12 | 0.5 |
| 63 | 84 | 42 | 4 | 2 | 14 | 12 | 0.5 |
| 64 | 67 | 34 | 4 | 2 | 14 | 12 | 0.5 |
| 65 | 200 | 100 | 1 | 3 | 3.5 | 20 | 0.5 |
| 66 | 125 | 63 | 1 | 3 | 3.5 | 20 | 0.5 |
| 67 | 84 | 42 | 1 | 3 | 3.5 | 20 | 0.5 |
| 68 | 67 | 34 | 1 | 3 | 3.5 | 20 | 0.5 |
| 69 | 200 | 100 | 2 | 3 | 7 | 20 | 0.5 |
| 70 | 125 | 63 | 2 | 3 | 7 | 20 | 0.5 |