# Peer review of "An automated fracture trace detection technique using the complex shearlet transform"

_Solid Earth, 2019_

## Short Comment (SC1) · 24 Jun 2019

This is a valuable and interesting contribution. Studies of outcropping reservoir successions - outcrop analogs - are a useful way to obtain distributed two- and three-dimensional rock data that are lacking in borehole-based observations and that encompass features below the resolution of seismic methods. Outcrops are thus a source of information on the likely attributes of fractures in the subsurface. This paper is an example of recent developments in rapid, automated image-based collection and analysis of fracture sizes, patterns and interconnections that are beginning to supply from outcrops valuable input for fracture models that go beyond fracture trace data painstakingly collected the old fashioned way.

[Figure]

I think, however, that it would not detract from the contribution presented in this paper, to mention a fundamental challenge facing remote/automatic extraction of fracture trace data from outcrop. If fractures are open or otherwise topographically prominent and make nice, detectable features, then all is well and only more efficient detection and extraction is needed. But open and topographically prominent may not be the case for many of the outcrop fractures that are the best subsurface analogs. Fractures that are open in in the subsurface are the ones we want to know about, for their effects on fluid flow and rocks strength, but fractures that become cemented shut in the subsurface may be provide the most reliable guides to subsurface patterns: these are 'fossilized' versions of the fractures were interested in. Such fractures are frequently the easiest to interpret as representative of the subsurface (e.g., can be separated from near-surface noise), they may by virtue of their fill history be the easiest to determine timing, origins, and to relate to specific targets in the subsurface (e.g., Ukar et al., 2019) and they commonly make the largest pavements (since there is no fracture porosity for plans to latch on to). But because they are filled, they are likely the least visible, or may be invisible, to remotish imaging.

Many of these issues are discussed with examples by Ukar et al. (2019).

Ukar, E., Laubach, S.E., Hooker, J.N., 2019. Outcrops as guides to subsurface natural fractures: example from the Nikanassin Formation tight-gas sandstone, Grande Cache, Alberta Foothills, Canada. Marine & Petroleum Geology, 103, 255-275. doi.org/10.1016/j.marpetgeo.2019.01.039

---

## Referee Comment (RC1) · Bertrand D.M. Gauthier (Referee) · 14 Aug 2019

**An automated fracture trace detection technique using the complex shearlet transform**

**General comments**

This paper describes a new method to automatically interpret fracture traces from 2D images (e.g. drone acquisition) using a novel technique. After explaining why an automated approach is better than manual interpretations, the method is briefly described in the main text and in more details in an appendix. The method is then applied to three different areas from different quality and resolution of images. Results are compared to manual interpretations. Finally advantages, disadvantages and way forward are discussed.

The paper is globally well organized, well written and easy to read despite some technical terms which are not clearly explained for the reader not familiar with these techniques. The figures are also globally well-presented although some of them would require some clarification. The abstract clearly summarizes the paper. A random check of the references did show any errors. The appendix is beyond my competence for a thorough review.

Considering the importance of such automatic interpretation methods and, from my knowledge, the original technique used, I accept this paper pending minor revision which are given below. My main concern is that if the (non-mathematician) reader can conclude that this shearlet transform allows convincing and fast automatic interpretation of fracture traces, he does not understand how physically it works.

**Introduction**

**Page 2 – line 2-6**

*Geomechanically derived DFNs are based on the physics of fracture propagation (Olson et al., 2009; Thomas et al., 2018) and can reproduce realistic fracture patterns* providing the complex paleostress field and paleo rock properties are known. *; however,* They are *also computationally intensive and hence have limited applicability. A carefully chosen fractured outcrop that is relatively free of noise (fractures resulting from exhumation and weathering* and not too much hidden by vegetation*) may be used to interpret realistic fracture networks which are geometrical inputs used in simulating various subsurface* thermo-hydro-mechanical-chemical processes (THMC) *processes.*

**Background**

**2.2 The Complex Shearlet Transform**

A shearlet definition for dummies (the simple geologists) and/or a simple analogy would be welcome in this chapter since it is the heart of the method. This chapter is reproduced from different references which are fundamentally mathematical hence difficult to understand for non-mathematician readers.

**Page 5 – line 11**

CoShREM with Canny, Sobel, phase congruency: ????

**Methods**

**3.2 Shearlet parameter selection**

The parameters which finally control the quality of the final fracture trace extraction are briefly described in Table 1 but their role and their physical meaning is not clear to me. Could it be possible to represent them on a figure (e.g. as part of Fig. 1).

**Page 6 - line 25**

*We use the structural similarity measure (SSIM)* : explain what it physically means or at least give a reference.

**Page 6 - line 29-30**

Mexican hat wavelet support: ???

Octave: ??? not defined even in Table 1.

**Results**

**Page 8 - Line 30**

*there is a tendency to interpret and link together disconnected features from the original raster image.*

Could it be possible to show differences in fracture length distribution between automatic and manual interpretation? (also valid for the other examples)

**Page 9 – line 4**

*(see Fig. 10a, 10b)*

**Page 9 – line 11**

*is shown in Fig. 10*

**Page 9 – line 16**

*Fig. 10d depicts the $P_{21}$*

**Page 9 – line 22**

*comparison between both  vectorizations*

**Page 9 – line 22**

*no real evidence of rock  failure*

**Page 10 – line 16**

*which are comparable in quality to the manual interpretation of Thiele et al. (2017) :* these manual interpretations are no shown so it is difficult for the reader to make his judgement.

**Discussion**

**Page 11 – line 10**

*King (2019), blob detection measures* : not clear what it is

**Page 12 – line 2-4**

K. Bisdom (2016) gives some relations between distance, resolution and camera length size which could be useful here (Burial related fracturing in sub-horizontal and fold reservoir – TU-Delft PhD thesis – ISBN 978-94-6186-740-7).

Since we are here in the suggestion part, you could also advise to make, if possible, 2 or 3 flight acquisitions at different altitudes to define resolution further.

**Page 12 – line 5-17**

The use of MPS is to mean important complement of the interpretation results. MPS could also be used to fill regions with false positives related to e.g. shrubbery.

**Appendix A**

This is beyond my competence. I cannot review this part.

**Figures**

**Figure 1**

Could be complemented by a drafted explanation of the shearlet transform parameters

**Figure 4**

In this present format, this figure does not mean anything for the basic reader. I suggest to shift it to the appendix and to replace it by the concrete effect of these parameters on the fracture trace extraction of a simple fracture network.

**Figure 5**

Could it be possible to add an image showing lineaments color coded as function of the relative number of time that they have been detected by each realization?

**Figure 8**

Figure 8b is not readable. I suggest 1) remove the photo underneath and 2) improve the contrast of the color scale (e.g. a three color legend bar scaled between 0 and >5 since it seems that there are very few zones above this threshold).

**Figure 9**

Again, try to improve the P21 color scale contrast

**Figure 10**

Same comment for 10d as for 8b

**Figure 11**

What is manual and what is automatic is not indicated. Put the fracture traces in white for better distinguishing them from the photo lineaments

**Figure 12**

    *(a) Bingie Bingie Area * 1

---

## Referee Comment (RC2) · Juliette Lamarche (Referee) · 10 Sep 2019

This paper presents a novel approach to provide accurate, exhaustive, realistic fracture patterns for performing DFN, which is valuable for using it in a variety of tectonic and lithologic settings. The paper is properly written and illustrated; the method is delivered in appendix. The manuscript is acceptable for publication with very few complements which are (1) noted along the text and (2) listed below. J.Lamarche Introduction The amount of literature on the topic is growing too fast at the moment. You cannot cite all of them, so please add "e.g." in your text. 4.1.2 Automated detection artificially fragments the large fractures, what are the methods, limitations, threshold values to prolongate traces and form large fractures? 4.1.3 Would be nice to have a better constrained comparison of automated versus visual extraction of fractures from French example.

In addition to the P21, could you provide data like strike (rose diagram or histograms), length (histogram), number of fractures. . . that will help understanding the nature of the difference between both visual and automated surveys. Would be nice to have them for all of the 3 field examples. Having a priori ideas on fractures while visually interpreting images is not a bad thing. Indeed, the geologist is aware of the brittle processes and their limitations on the fracture geometries. This, to my opinion, is precious for visual "human-hand" fracture tracing. 4.3: No difference is made between geological features such as fractures, bedding, other, when automated tracking is performed. So, the need to impose a priori structures or preferred fracture trends is important upstream and worth in order to avoid interpreting excessive or ghost fractures. 5. Still the edges are detected when sharp on the topography. This is the condition for exhaustive tracking and possible comparison between world-wide remote outcrops. Pecularily, in carbonates long-lasting aerial exposure alters the colors, softens the fracture crests and smooth the reliefs. This is sometimes -but not always- related to karst genesis. A discussion on the limitations bias resulting from exposure conditions could be welcome. FIGURES & CAPTIONS Fig. 11: indicate which one, between left and right, is manual and automated -

---

## Author Comment (AC1) · 13 Oct 2019

Reviewer Comment - General This is a valuable and interesting contribution. Studies of outcropping reservoir successions - outcrop analogs - are a useful way to obtain distributed two- and threedimensional rock data that are lacking in borehole-based observations and that encompass features below the resolution of seismic methods. Outcrops are thus a source of information on the likely attributes of fractures in the subsurface. This paper is an example of recent developments in rapid, automated image-based collection and analysis of fracture sizes, patterns and interconnections that are beginning to supply from outcrops valuable input for fracture models that go beyond fracture trace data painstakingly collected the old fashioned way.

[Figure]

Author's Response to General Comment The authors would like to thank Dr. Laubach for the comment. We agree that photogrammetric outcrop datasets are highly valuable and there is a need for tools to efficiently process and deliver insights into fracture patterns from outcrop fracture data so that they complement hard data from boreholes.

Reviewer Comment #1 I think, however, that it would not detract from the contribution presented in this paper, to mention a fundamental challenge facing remote/automatic extraction of fracture trace data from outcrop. If fractures are open or otherwise to-pographically prominent and make nice, detectable features, then all is well and only more efficient detection and extraction is needed. But open and topographically promi-nent may not be the case for many of the outcrop fractures that are the best subsurface analogs. Fractures that are open in in the subsurface are the ones we want to know about, for their effects on fluid flow and rocks strength, but fractures that become ce-mented shut in the subsurface may be provide the most reliable guides to subsurface patterns: these are 'fossilized' versions of the fractures were interested in. Such frac-tures are frequently the easiest to interpret as representative of the subsurface (e.g., can be separated from near-surface noise), they may by virtue of their fill history be the easiest to determine timing, origins, and to relate to specific targets in the subsurface (e.g., Ukar et al., 2019) and they commonly make the largest pavements (since there is no fracture porosity for plans to latch on to). But because they are filled, they are likely the least visible, or may be invisible, to remotish imaging. Many of these issues are discussed with examples by Ukar et al. (2019).

Author's Response to Reviewer Comment #1 We agree with the reviewer that detailed investigation (using fluid inclusion studies, SEM-CL imaging etc.) into fracture cement infill can provide a more clearer picture into the evolution, timing, and stress history of fractures. Therefore, the guidelines presented by Ukar et al, 2019 on choosing out-crops representative of subsurface conditions has significant merit. This reference is added within the Introduction section of the marked down manuscript so that the reader is aware that not all outcrop fracture data can be readily extrapolated to subsurface

conditions. In our case, the UAV data acquisition in France and Brazil was performed, not with a specific goal to extrapolate any observed outcrop fracturing patterns to a known sub-surface reservoir, but for a broader perspective of deformation in different carbonate settings.

The issue of identifying, on an outcrop scale, mineralized fractures is indeed a pertinent one. Extracting filled opening mode fractures, in which mineral fill has a marked colour contrast with respect to host rock, can be done with segmentation algorithms if the images have been acquired in close range. The complex shearlet transform could also be successful in such a case, provided this contrast is prominent. The photogrammetric datasets that we present in the manuscript, were unfortunately acquired at altitudes from which these filled macrofractures were below the necessary image spatial resolution. We would suggest close range UAV mounted hyperspectral imaging as a possible technique to extract opening mode fractures with cement infill. With hyperspectral imaging (or imaging spectroscopy), image data is collected in near-continuous spectral bands. The spectral response of minerals constituting the rock, owes to atomic-molecular level processes triggered on interaction with a light source (active or passive) and this may be utilized to identify mineral composition. Since mineral fill of veins are likely to have a different spectral response from the mineralogy of the host rock, this variation may be used to isolate the pixels that correspond to veins. A recent review on close range hyperspectral imaging for mineral identification identifies various previous studies performed for specific minerals (Krupnik and Khan, 2019) with spectral response bands for the most common mineral types. It would be interesting to observe, identify, and distinguish between fracture sequences (at least for the macrofractures) based on the differences in spectral response of the fracture infill material. Since hyperspectral data is much more voluminous and with significantly more complex image processing than conventional photogrammetry, such analysis could be confined to selected regions within the outcrop. Together with conventional UAV photogrammetry that covers larger spatial area, laboratory based geochemical studies, and outcrop observations (scanline sampling, abutting relations etc.), hyperspectral

methods can yield a more detailed picture of a particular outcrop setting.

References Krupnik D, Khan S (2019), Close-range, ground-based hyperspectral imaging for mining applications at various scales: Review and case studies, Earth-Science Reviews, In Press, https://doi.org/10.1016/j.earscirev.2019.102952

Ukar, E., Laubach, S.E., Hooker, J.N., 2019. Outcrops as guides to subsurface natural fractures: example from the Nikanassin Formation tight-gas sandstone, Grande Cache, Alberta Foothills, Canada. Marine & Petroleum Geology, 103, 255-275. https://doi.org/10.1016/j.marpetgeo.2019.01.039

---

## Author Comment (AC2) · 13 Oct 2019

The authors would like to thank Dr. Lamarche for the reviewer comments. Please note our responses to the specific comments below.

Reviewer Comment #1 Introduction. The amount of literature on the topic is growing too fast at the moment. You cannot cite all of them, so please add "e.g." in your text.

Author's Response to Reviewer Comment #1 We have modified the particular citations within the Introduction section of the marked down manuscript.

Reviewer Comment #2 4.1.2 Automated detection artificially fragments the large fractures, what are the methods, limitations, threshold values to prolongate traces and form large fractures? Printer-friendly version

Author's Response to Reviewer Comment #2 This is a very valid comment. Automated techniques can result in fragmentation with and without gaps. If the fragmentation happens without a gap, then the sum of lengths of individual segments that constitute a single trace is equal to full trace length. Our implementation fits polylines to intersecting pixel clusters (representing intersecting fractures), inserts a branch point, and stores the polylines separately. We do not see this kind of fragmentation as an issue when such a fragmented DFN is used for geometric input for flow / geomechanics simulation. This is because the process of meshing models with explicitly specified DFN geometry would in any case require the specification of all intersection points (or the forced fragmentation of long fractures). If an interpreter wants to identify certain segments as belonging to one fracture set, regardless of the number of intersections, then more information is needed to arrive at such a decision. This information could be based on orientation, abutting relationships, mineralogical study of cement infill, and fluid inclusion studies. Fracture strike is easily calculated (since each polyline is georeferenced). To link fragmented traces based on strike, we could implement simple heuristics. In case of fragments that share an intersection point (or branch point), we could parse through the stored list of fractures, identify those with similar strike and enforce them as one fracture set. However, strike alone may not be enough to identify whether fragments are to be joined as a single set.

A single fracture (as interpreted by the geologist) could also be fragmented without being cut by other intersecting fractures. This could happen in the case of false negatives (shadows or shrubbery within an open fracture) that cause fragmentation of fractures with gaps in between them. This kind of fragmentation affects the topology of the network. The role of fracture network topology on fluid flow response has been the focus of many works such (for example Hardebol et al, 2015). The simplest strategy in this case would be to use a linking threshold plus similar strike to bridge these gaps and store the fragments as a single fracture. Such approaches are already available (such as NetworkGT by Nyberg et al, 2018). The uncertainty in topologies (in terms of the relative I-Y-X nodes within the network) maybe studied to further quantify simulation
responses. However, these methods may not generalize well in all cases, especially when the fragmentation is due to noisy data.

Within the machine learning literature, there has been some progress on deep graph algorithms that can perform graph extrapolation (Cordonnier and Loukas, 2019). These techniques have been developed based on the premise that naturally occurring graph structures are almost always, incompletely sampled. Therefore, natural networks would require manual intervention to add edges or nodes based on the relevant domain expertise. The edges may need to be extrapolated from a set of training graphs (that encapsulate features of a complete natural network) which can then extrapolate edges to an incompletely sampled graph using generative graph neural networks. The application of such techniques is beyond the scope of this manuscript but we will attempt this in future work. Collected field observations such as Lamarche et al, 2018 that specify dimensional thresholds for fracture linkage could be useful for such an attempt.

Reviewer Comment #3 4.1.3 Would be nice to have a better constrained comparison of automated versus visual extraction of fractures from French example. In addition to the P21, could you provide data like strike (rose diagram or histograms), length (histogram), number of fractures. . . that will help understanding the nature of the difference between both visual and automated surveys. Would be nice to have them for all of the 3 field examples.

Author's Response to Reviewer Comment #3 We have added rose diagrams, cumulative length distributions, and number of traces to both manual and automatic interpretations corresponding to the Parmelan (Fig.1(f) and Fig.1(g)) and Brejoes examples (see Fig.2 for the comparison and Fig.3 for the rose plots and cumulative distributions). Please note that for the Bingie Bingie examples, Thiele et al, 2017 have not released vectorised assisted interpretation trace maps and hence we created vectorised trace maps from the raster images presented in Thiele et al, 2017 (see Fig.4(k) and Fig.5(k)). This additional plot is used only for visual comparison with Fig.4(h) and Fig.5(h). To
Author's Response to Reviewer Comment #5 We have added the following sentences within the Discussion section (under Detection of large cavities and false positives) in the marked down manuscript to address this issue, "Additionally, carbonate outcrops

Reviewer Comment #5 5. Still the edges are detected when sharp on the topography. This is the condition for exhaustive tracking and possible comparison between worldwide remote outcrops. Pecularily, in carbonates long-lasting aerial exposure alters the colors, softens the fracture crests and smooth the reliefs. This is sometimes -but not always- related to karst genesis. A discussion on the limitations bias resulting from exposure conditions could be welcome.

Author's Response to Reviewer Comment #4 We agree that a priori ideas are, in general, useful in performing manual tracing. In Page 8, Lines 29-31, we mentioned this to highlight the fact that an interpreter who hand traces at a fixed zoom level, would perhaps not identify disconnected fragments unlike automatic interpretation which works on original image resolution. It is true that the ridge detection cannot distinguish between fracture, bedding planes, shadows etc. A priori knowledge can indeed help in removing false positives.

4.3: No difference is made between geological features such as fractures, bedding, other, when automated tracking is performed. So, the need to impose a priori structures or preferred fracture trends is important upstream and worth in order to avoid interpreting excessive or ghost fractures.

Reviewer Comment #4 Having a priori ideas on fractures while visually interpreting images is not a bad thing. Indeed, the geologist is aware of the brittle processes and their limitations on the fracture geometries. This, to my opinion, is precious for visual "human-hand" fracture tracing.

quantitatively compare between automatic and assisted interpretations, rose plots are added in Fig.4(f), Fig.4(i), Fig.5(f), and Fig.5(i). The cumulative length distributions are added in Fig.4(g), Fig.4(j), Fig.5(g), and Fig.5(j).
are prone to widespread erosion owing to exposure to meteoric water from precipitation cycles and air corrosion. Geomorphological features owing to these effects may also play a role in generation of false positives."

Reviewer Comment #6 Fig. 11: indicate which one, between left and right, is manual and automated –

Author's Response We have added this detail to the Figure. The figure caption will be modified ("...automatic and manual.." changed to "...manual and automatic") to avoid confusion to the reader.

Cordonnier, J-B nad Loukas A (2019), Extrapolating Paths with Graph Neural Networks, CoRR, abs/1903.07518, http://arxiv.org/abs/1903.07518

Hardebol NJ, Maier C, Nick H, Geiger S, Bertotti G, Boro H (2015), Multiscale fracture network characterization and impact on flow: A case study on the Latemar carbonate platform, Journal of Geophysical Research - Solid Earth, 120(12), https://doi.org/10.1002/2015JB011879

Lamarche J, Chabani A, Gauther BDM (2018), Dimensional threshold for fracture linking and hooking, Journal of Structural Geology, Vol 108, 171-179, https://doi.org/10.1016/j.jsg.2017.11.016 SED

---

## Editor Comment (EC1) · Federico Rossetti (Editor) · 14 Oct 2019

Dear Authors,

based on the reviewers' reports and the way you responded to their comments/remarks, you are invited to submit a revised version of your manuscript.

My assessment of the submitted manuscript is in line with the referees' comments when requiring (i) a more exhaustive explanation of the adopted methodology; (ii) a better comparison between the structural data collected in the field and the output results; and (iii) an explicit sensitivity analysis for a better understanding of the relationships between input and output variables.

Regarding points (ii) and (iii), I would ask the Authors to include in the Discussion

chapter a section entitled "Validity and Limitation of the Technique", where advantages and disadvantages of the adopted technical approach are systematically listed and critically discussed (see also the SC1 on this regard). This addition is necessary for the overall appraisal of the obtained results and their impact for future research in the field.

Specific comments (i) Figure 2: it merits a more exhaustive description both in the text and figure caption (i) Figures 8 and 10-13: include the number of measurements in rose diagrams. As a general comment, rose diagrams are adequate in depicting the strike of sub-vertical joint arrays. Have the measured joint systems sub-vertical attitudes?

Yours sincerely, Federico Rossetti
* * *

---

## Author Comment (AC3) · 14 Oct 2019

Author's Response The authors would like to thank the Dr. Gauthier for his comments. Please see the attached text file that details the reply to specific suggestions and modifications made. The marked down manuscript created using latexdiff highlights the changes made. Since a significant number of comments were related to the figures, we summarize the changes made to the figures.

Figures that are modified:

Figure 1 (in old manuscript): Inserted as an appendix figure. Referenced within the text of the Appendix. (In the marked down manuscript, referred to as Fig.A1)

Figure 4 (in old manuscript: Moved to the appendix. Referenced within Section

[Figure]

3.2 where the parameter selection procedure is explained. (In the marked down manuscript, referred to as Fig.A3)

Figure 8 (in old manuscript): The map behind the Parmelan P21 plots in Fig.8(b) is removed and the colorbar scaling has been modified. (In the marked down manuscript, referred to as Fig.7)

Figure 9 (in old manuscript): The colorbar scaling has been modified for the Parmelan manual and automatic interpretations. Rose plots and cumulative frequency plots are added. (In the marked down manuscript, referred to as Fig.8)

Figure 10 (in old manuscript): The colorbar scaling has been modified for the Brejoes P21 plots. (In the marked down manuscript, referred to as Fig.9)

Figure 11 (in old manuscript): The caption has been modified to properly indicate which is manual and automatic. Trace color changed from red to white for better visualization. (In the marked down manuscript, referred to as Fig.10)

Figure 12 (in old manuscript): Assisted trace interpretation results from the original paper of Thiele et al, 2017 has been added for comparison. Rose plots and cumulative length distributions added. (In the marked down manuscript, referred to as Fig.12)

Figure 13 (in old manuscript): Assisted trace interpretation results from the original paper of Thiele et al, 2017 has been added for comparison. Rose plots and cumulative length distributions added. (In the marked down manuscript, referred to as Fig.13)

Figures that have been newly added:

Figure 2 (in marked down manuscript): Depiction of ridge extraction results that vary depending upon parameter changes. This is applied to a very simple fractured sandstone image

Figure 11 has been inserted that depicts rose plots and cumulative frequency distributions for all the stations

Please also note the supplement to this comment:
https://www.solid-earth-discuss.net/se-2019-104/se-2019-104-AC3-supplement.pdf

**Supplement:**

Reply to Review by Dr. Bertrand Gauthier

Reviewer Comment

**General comments**

This paper describes a new method to automatically interpret fracture traces from 2D images (e.g. drone acquisition) using a novel technique. After explaining why an automated approach is better than manual interpretations, the method is briefly described in the main text and in more details in an appendix. The method is then applied to three different areas from different quality and resolution of images. Results are compared to manual interpretations. Finally advantages, disadvantages and way forward are discussed.

The paper is globally well organized, well written and easy to read despite some technical terms which are not clearly explained for the reader not familiar with these techniques. The figures are also globally well-presented although some of them would require some clarification. The abstract clearly summarizes the paper. A random check of the references did show any errors. The appendix is beyond my competence for a thorough review.

Considering the importance of such automatic interpretation methods and, from my knowledge, the original technique used, I accept this paper pending minor revision which are given below. My main concern is that if the (non-mathematician) reader can conclude that this shearlet transform allows convincing and fast automatic interpretation of fracture traces, he does not understand how physically it works.

Author's Response

The authors would like to thank the reviewer for his comments. We have added a new figure within the background section that highlights the changes in detection parameters on the extraction results.

Reviewer Comment #1

**Page 2 – line 2-6**

*Geomechanically derived DFNs are based on the physics of fracture propagation (Olson et al., 2009; Thomas et al., 2018) and can reproduce realistic fracture patterns* providing the complex paleostress field and paleo rock properties are known. *; however, They are* also *computationally intensive and hence have limited applicability. A carefully chosen fractured outcrop that is relatively free of noise (fractures resulting from exhumation and weathering* and not too much hidden by vegetation*) may be used to interpret realistic fracture networks which are geometrical inputs used in simulating various subsurface* thermo-hydro-mechanical-chemical processes (THMC) *processes.*

Author's Response to Reviewer Comment #1

We have modified the text incorporating the suggested sentence corrections by the reviewer (See Page 2, Lines 4-7 in the marked down manuscript).

Reviewer Comment #2

**2.2 The Complex Shearlet Transform**

A shearlet definition for dummies (the simple geologists) and/or a simple analogy would be welcome in this chapter since it is the heart of the method. This chapter is reproduced from

different references which are fundamentally mathematical hence difficult to understand for non-mathematician readers.

**Author's Response to Reviewer Comment #2**

We will modify the text in this section in the revised manuscript for a more simpler explanation behind the theory of the complex shearlet transform.

**Reviewer Comment #3**

**Page 5 – line 11**
CoShREM with Canny, Sobel, phase congruency: ????

**Author's Response to Reviewer Comment #3**

This sentence was quite unclear and is replaced with "..complex shearlet based feature detection compared with conventional edge detectors such as Canny (Canny, 1986), Sobel (Sobel and Feldman, 1973), phase congruency (Kovesi, 1999) ..." in the revised manuscript. (See Page 5, Lines 19-21 in the marked down manuscript).

**Reviewer Comment #4**

**3.2 Shearlet parameter selection**
The parameters which finally control the quality of the final fracture trace extraction are briefly described in Table 1 but their role and their physical meaning is not clear to me. Could it be possible to represent them on a figure (e.g. as part of Fig. 1).

**Author's Response to Reviewer Comment #4**

We have added a new figure (see Figure 2, Page 26 in the marked down manuscript) referenced within Section 3.1 (see Page 4, Lines 7-8 in the marked down manuscript) where the effect of parameters are highlighted. A simple example of fractured sandstone rock is chosen to depict to highlight the effects.

**Reviewer Comment #5**

**Page 6 - line 25**
*We use the structural similarity measure (SSIM)* : explain what it physically means or at least give a reference.

**Author's Response to Reviewer Comment #5**

The reference to the original paper (Wang et al., 2004) that introduced the SSIM was already in the manuscript (Page 6, Line 27 in old manuscript). We have now moved this reference to the first instance where the SSIM is referred to in the manuscript (See Page 7, Lines 5-6 in the marked down manuscript)

**Reviewer Comment #6**

**Page 8 - Line 30**
*there is a tendency to interpret and link together disconnected features from the original raster image.*
Could it be possible to show differences in fracture length distribution between automatic and manual interpretation? (also valid for the other examples)

**Author's Response to Reviewer Comment #6**

Length weighted rose diagrams and cumulative trace length distributions are added to the figures where automatic and manual interpretations are depicted for Parmelan and Brejoes examples. In the case of Bingie Bingie examples rose plots and cumulative trace length distributions are added for the automatic and assisted interpretations (see Figures 8, 11, 12, and 13 in the marked down manuscript).

**Reviewer Comment #7**

**Page 9 – line 4**
*(see Fig. 10a, 10b)*

**Author's Response to Reviewer Comment #7**

Modified within revised manuscript (see Page 9, Line 15 in the marked down manuscript. The figure number has changed to Fig.9).

**Reviewer Comment #8**

**Page 9 – line 11**
*is shown in Fig. 10c*

**Author's Response to Reviewer Comment #8**

Modified within revised manuscript (see Page 9, Line 22 in the marked down manuscript. The figure number has changed to Fig.9c).

**Reviewer Comment #9**

**Page 9 – line 16**
*Fig. 10d depicts the $P_{21}$*

**Author's Response to Reviewer Comment #9**

Modified within revised manuscript (see Page 9, Line 27 in the marked down manuscript. The figure number has changed to Fig.9).

**Reviewer Comment #10**

**Page 9 – line 22**
*comparison between both  vectorizations*

**Author's Response to Reviewer Comment #10**

Modified within revised manuscript (see Page 10, Line 4 in the marked down manuscript).

**Reviewer Comment #11**

**Page 9 – line 22**
*no real evidence of rock  failure*

**Author's Response to Reviewer Comment #11**

Modified within revised manuscript (see Page 10, Line 7 in the marked down manuscript).

**Reviewer Comment #12**

**Page 10 – line 16**

*which are comparable in quality to the manual interpretation of Thiele et al. (2017) :* these manual interpretations are no shown so it is difficult for the reader to make his judgement.

**Author's Response to Reviewer Comment #12**

Thiele et al, 2017 have not released vectorised versions of their assisted interpretations. Hence, we have interpreted a trace map derived from the assisted interpretations in that paper and added to the relevant figures of Bingie Bingie #1 and #2 (see Fig.12(k) and Fig.13(k) in the marked down manuscript).

**Reviewer Comment #13**

**Page 11 – line 10**
*King (2019), blob detection measures* : not clear what it is

**Author's Response to Reviewer Comment #13**

The pre-print of Reisenhofer and King (2019) has now been published in SIAM Journal on Imaging Sciences and we have updated the reference. A blob within an image, is simply a group of connected pixels that differ in properties as compared to the surrounding. Compared to thin features such as edges and ridges which are akin to lower dimensional structures within a 2D image, blobs are 2D regions within the same 2D image. In our carbonate examples, weathering has caused such cavities which are better extracted using blob detection algorithms rather than ridge detection measures hence we referred to these techniques.

**Reviewer Comment #14**

**Page 12 – line 2-4**
K. Bisdom (2016) gives some relations between distance, resolution and camera length size which could be useful here (Burial related fracturing in sub-horizontal and fold reservoir – TU-Delft PhD thesis – ISBN 978-94-6186-740-7).

Since we are here in the suggestion part, you could also advise to make, if possible, 2 or 3 flight acquisitions at different altitudes to define resolution further.

**Author's Response to Reviewer Comment #14**

The relation provided by Bisdom, 2016 connecting resolution to flight altitude is specific to the type of camera that was used i.e., OPTIO S1. Multiple flights at same location to define the ideal resolution is a good suggestion and we have added the following sentence to the second last paragraph of the discussion. "The ideal flying resolution to identify features of interest may be ascertained by carrying out a series of acquisitions at a location where ground truth is known." (see Page 12, Lines 20-21 in the marked down manuscript)

**Reviewer Comment #15**

**Page 12 – line 5-17**
The use of MPS is to mean important complement of the interpretation results. MPS could also be used to fill regions with false positives related to e.g. shrubbery.

**Author's Response to Reviewer Comment #15**

Our suggestion was to use the method to generate realistic training image libraries for MPS based DFN generation. A similar exercise was performed by Pyrcz et al, 2008 for fluvial and deepwater systems. Such a library of training images needs to be carefully constructed after ensuring removal of any false

positives. Failure to do so will replicate these false features in the output of the MPS algorithm. We have modified sentence in Page 12, Lines 10-11 "Our automated method can quickly produce accurate, geologically realistic, and unbiased training images that can feed into the MPS workflow" into "Corrected for false positives and noise, the automated method can produce accurate, geologically realistic, and unbiased training images that can feed into the MPS workflow." (see Page 12, Lines 27-29 in the marked down manuscript)

**Reviewer Comment #16**

**Figure 1**

Could be complemented by a drafted explanation of the shearlet transform parameters

**Author's Response to Reviewer Comment #16)**

We have added a new figure within Section 3.1 depicting the variation in each parameter on a simple example from a siliciclastic fractured rock sample (see Figure 2 in the marked down manuscript).

**Reviewer Comment #18**

**Figure 4**

In this present format, this figure does not mean anything for the basic reader. I suggest to shift it to the appendix and to replace it by the concrete effect of these parameters on the fracture trace extraction of a simple fracture network.

**Author's Response to Reviewer Comment #18**

Figure 4 is moved to the Appendix. The new figure depicting effect of parameters is placed within Section 3.1 (see Figure 2 in the marked down manuscript).

**Reviewer Comment #19**

**Figure 5**

Could it be possible to add an image showing lineaments color coded as function of the relative number of time that they have been detected by each realization?

**Author's Response to Reviewer Comment #19**

This is complicated to do, as when the vectorization is performed for each realization, there could be a slight shift in the position of the lineaments. The lineament lengths may also change in every realization making a one-to-one comparison quite difficult. However, in the revised manuscript we could make an image that displays the final trace map overlaid on a pixelated image that is colour coded with the number of times the particular pixel attained a non-zero intensity.

**Reviewer Comment #20**

**Figure 8**

Figure 8b is not readable. I suggest 1) remove the photo underneath and 2) improve the contrast of the color scale (e.g. a three color legend bar scaled between 0 and >5 since it seems that there are very few zones above this threshold).

**Author's Response to Reviewer Comment #20**

We have removed the underlying photograph and modified the colour bar scaling to better emphasize the variation in $P_{21}$ (see Figure 7 in the marked down manuscript).

Reviewer Comment #21

**Figure 9**

Again, try to improve the P21 color scale contrast

Author's Response to Reviewer Comment #21

As per reviewers suggestion, we have modified this figure with better scaling of colour bar (see Figure 8 in the marked down manuscript).

Reviewer Comment #22

**Figure 10**

Same comment for 10d as for 8b

Author's Response to Reviewer Comment #22

As per reviewers suggestion, we have modified figure with better scaling of colour bar (see Figure 9 in the marked down manuscript).

Reviewer Comment #23

**Figure 11**

What is manual and what is automatic is not indicated. Put the fracture traces in white for better distinguishing them from the photo lineaments

Author's Response to Reviewer Comment #23

We have modified the figure 11 with the fracture traces in white colour. Caption is modified to identify the manual and automatic interpretations (see Figure 10 in the marked down manuscript).

Reviewer Comment #24

**Figure 12**

*(a) Bingie Bingie Area  1*

Author's Response to Reviewer Comment #24

We have applied the above corrections to the revised manuscript (see Figure 12 in the marked down manuscript).

**References**

Pyrcz MJ, Boisvert JB, Deutsch C (2008), A library of training images for fluvial and deepwater reservoirs and associated code, Computers and Geosciences, 34(5):542-560, 10.1016/j.cageo.2007.05.015

Thiele, S. T., Grose, L., Samsu, A., Micklethwaite, S., Vollgger, S. A., and Cruden, A. R.: Rapid, semi-automatic fracture and contact mapping for point clouds, images and geophysical data, Solid Earth, 8 (6), 1241–1253, https://doi.org/10.5194/se-8-1241-2017

Reisenhofer R and King EJ (2019), Edge, Ridge and Blob Detection with Symmetric Molecules, SIAM Journal of Imaging Sciences, 12(4), 1585-1626, , https://doi.org/10.1137/19M1240861

---

## Author Comment (AC4) · 14 Oct 2019

[revised manuscript text omitted]
 to overcome the shortcoming of wavelets by applying dilation, shear transformation and translation operations to wavelet generating functions. Shearlets are hence very similar to wavelets except that isotropic dilation of wavelets is replaced with anisotropic dilation

5 and shearing(see Fig. A1a, Fig. A1b)... Shearlets have a number of properties that make them better suited to handle sparse, geometric features in multidimensional data compared to traditional wavelets (Kutyniok and Labate, 2012).

The complex shearlet transform is a complex-valued generalization of the shearlet transform that was developed by Labate et al. (2005) to handle geometric structures in 2D data. Reisenhofer (2014) and King et al. (2015) proposed the idea of creating

- 10 complex shearlets by modifying the shearlet construction so that real parts of the generating function are even-symmetric and imaginary parts of the generating function is odd-symmetric. They used the Hilbert transform to convert an even-symmetric function into an odd-symmetric function and vice versa. The complex shearlet measure for ridge and edge detection implemented in Reisenhofer (2014); King et al. (2015) and Reisenhofer et al. (2016) merged the ideas of phase congruency and complex shearlets.
- 15

The complex shearlet measure first introduced by Reisenhofer (2014) and improved by King et al. (2015) was used for applications like coastline detection King et al. (2015), flame front detection Reisenhofer et al. (2016), and feature extraction from terrestrial LIDAR inside tunnels Bolkas et al. (2018). Karbalaali et al. (2018) used the complex shearlet transform for channel edge detection from synthetic and real seismic slices. Reisenhofer et al. (2016) presented a comprehensive comparison

20 of CoShREM with Cannycomplex shearlet based feature detection compared with conventional edge detectors such as Canny (Canny, 1986), Sobel (Sobel and Feldman, 1973), Sobel, phase congruency (Kovesi, 1999), and another shearlet based edge detector Yi et al. (2009) (Yi et al., 2009). Bolkas et al. (2018) also makes made specific comparisons between the performance of Canny, Sobel, Prewitt (Prewitt, 1970) edge detection methods versus space-frequency transform methods such as wavelets, contourlets, and shearlets. A detailed overview of the complex shearlet transform is provided in Appendix. A for the interested reader.

**3 Methods**

**3.1 The Automatic Detection Process**

The automated fracture trace detection method that we present has five main steps (see Fig. 1). The first step of the method uses the Complex Shearlet-Based Ridge and Edge Measure (CoShREM), a MATLAB implementation by Reisenhofer et al. (2016).

30 The first step, namely the ridge detection, is dependent on a number of input parameters tabulated in Table 1 and Table 2. Equation (A28) gives the expression for the ridge measure. An optimal set of deterministic parameter values which can extract features on all scales is not known a priori. Therefore, we vary the input parameters corresponding to the construction of the shearlet system and the ridge detection parameters within user-defined ranges to compute multiple ridge realizations. A ridge ensemble map is obtained by superposing the ridge images and normalizing. A user-defined threshold is then applied to the intensity values of the normalized ride ensemble image to extract a highly probable, binarized, ridge network. The threshold is set by a visual comparison of the input image with the extracted ridges. The range for each parameter in Table 1 and Table 2

- 5 is ascertained by first testing the effect of variation of each parameter with respect to a chosen base case image. This approach to automated detection captures features of multiple scales and highlights regions of uncertain feature extraction within the image. An example of a fractured image and the effect of parameter variation on ridges that are detected is depicted in Fig.2. The sample represents 
[revised manuscript text omitted]

---

## Author Comment (AC5) · 14 Oct 2019

Please find attached marked down manuscript with changes.

Please also note the supplement to this comment:
https://www.solid-earth-discuss.net/se-2019-104/se-2019-104-AC5-supplement.pdf

—————————————————

---

## Author Response (AR1)

Rahul Prabhakaran,
Corresponding Author,
Manuscript SE-2019-104

November 8, 2019

Dr. Federico Rossetti
Handling Editor - Manuscript SE-2019-104
Solid Earth Journal
Copernicus Publications

Dear Dr. Rossetti

We thank you for your valuable comments and suggestions to our reply to reviewers and short comments.

Please find attached a revised manuscript as well as a marked down version that highlights the changes w.r.t the last version (submitted on 14.10.2019). The main highlights of revisions within the new manuscript are the following:

- **Explanation of complex shearlet parameters, their effects, and sensitivity analysis** We have added a new sub-section within Methods, referred to as 3.2 Sensitivity Analysis of Parameters on Extraction Results. Alongwith minor additions to Section 2.2 The Complex Shearlet Transform, Section 3.2 presents a detailed explanation on the parameters and their effects on trace extraction applied to the simple siliciclastic rock fracture example (previously Figure 2, now Figure 3). Figure 3 has also been improved by adding respective vectorized trace result adjacent to each ridge map, so that the reader can judge the effect of each parameter in terms of vectorized traces. An additional figure is also added that showcases the effects of parameters on the complex shearlet construction (Figure 4 in the new manuscript).

- **Detection of veins** We fully agree with by Dr. Stephen Laubach on the importance of mineralized fractures in structural studies and, indeed for the behaviour of fractured reservoirs. Our contribution is geared to the detection of lineaments visible from the camera of a drone. In most cases, these are open fractures. The mineralized fractures in Parmelan have an infill that is not very prominent. We have provided an additional example (handheld camera) within the Parmelan results (Section 4.1.4, Figure 10 in revised manuscript) that indicate that extracting filled fractures is possible and therefore do not constitute a limitation.

- **Structural Data** The structural data on mineralized fractures that we have collected in the field confirm that the attitude of fractures are predominantly vertical. This is also the case for open fractures that are

observed from photogrammetry and which we have confirmed in the field. Orientations of automatically extracted fractures and ground truth structural data match in both the Brazilian and French examples. Structural measurements are added to relevant figures of both Parmelan (Figure 8 in the new manuscript) and Brejões (Figure 11 in the new manuscript).

- **Validity and Limitations** The discussion section is expanded with a section where issues of vein extracton, false positives, and artificial fragmentation are critically discussed. These issues constituted our Reply to Reviewers (and Short Comments) and are now within the revised manuscript.

- **Change in Acknowledgement** On request of Dr. Hilario Bezerra, the acknowledgements to Shell and ANP are removed from the acknowledgements section. This was because the sponsors of the Brejões fieldwork were already properly acknowledged in the publication by Boersma er al, 2019 and we only use data that was permitted to be released.

On Behalf of co-authors, Dr. Pierre-Olivier Bruna, Prof.Dr. Giovanni Bertotti, Prof.Dr.ir David Smeulders

Rahul Prabhakaran, PhD Candidate, Section of Applied Geology, TU Delft

[revised manuscript text omitted]